**Subject Area:**
structural biology/cellular biology/molecular biology

centromere, kinetochore, mitosis, chromatin, epigenetics, nucleosome

**Author for correspondence:**
Ben E. Black
e-mail: blackbe@pennmedicine.upenn.edu

# The centromere comes into focus: from CENP-A nucleosomes to kinetochore connections with the spindle

Kathryn Kixmoeller[1,2,3,4], Praveen Kumar Allu[1,2,3] and Ben E. Black[1,2,3,4]

[1]Department of Biochemistry and Biophysics, [2]Penn Center for Genome Integrity, [3]Epigenetics Institute, and [4]Graduate Program in Biochemistry and Molecular Biophysics, Perelman School of Medicine, University of Pennsylvania, Philadelphia, PA 19104, USA

KK, 0000-0002-7996-0279; PKA, 0000-0001-7547-2254; BEB, 0000-0002-3707-9483

Eukaryotic chromosome segregation relies upon specific connections from DNA to the microtubule-based spindle that forms at cell division. The chromosomal locus that directs this process is the centromere, where a structure called the kinetochore forms upon entry into mitosis. Recent crystallography and single-particle electron microscopy have provided unprecedented high-resolution views of the molecular complexes involved in this process. The centromere is epigenetically specified by nucleosomes harbouring a histone H3 variant, CENP-A, and we review recent progress on how it differentiates centromeric chromatin from the rest of the chromosome, the biochemical pathway that mediates its assembly and how two non-histone components of the centromere specifically recognize CENP-A nucleosomes. The core centromeric nucleosome complex (CCNC) is required to recruit a 16-subunit complex termed the constitutive centromere associated network (CCAN), and we highlight recent structures reported of the budding yeast CCAN. Finally, the structures of multiple modular sub-complexes of the kinetochore have been solved at near-atomic resolution, providing insight into how connections are made to the CCAN on one end and to the spindle microtubules on the other. One can now build molecular models from the DNA through to the physical connections to microtubules.

## 1. Introduction

Ensuring the faithful propagation of genetic information across generations is one of the most fundamental problems of cell biology. Each chromosome must be properly replicated and segregated during every cell cycle [1]. At mitosis, each pair of sister chromatids must align along the metaphase plate and form proper attachments to spindle microtubules such that the sister chromatids segregate towards opposite poles, leaving one chromatid in each resultant daughter cell [2]. The crucial connection to spindle microtubules occurs at a region of the chromosome called the centromere and is mediated by a large protein assembly called the kinetochore [3–6]. The kinetochore assembles at the centromere and spans the distance between centromeric chromatin and spindle microtubules. The kinetochore serves many functions including ensuring proper bi-orientation of sister chromatids (in mitosis) or bivalents (in meiosis), stabilizing kinetochore–microtubule attachments, and preventing incorrect attachments that could lead to erroneous chromosome segregation [2,7].

The first identification of kinetochore proteins occurred when sera obtained from patients with the autoimmune disease scleroderma were found to contain autoantibodies against the centromere region of chromosomes [8]. The targets of these autoantibodies were three centromeric proteins, named CENP (*CEN*tromeric *P*rotein)-A, CENP-B and CENP-C [9]. Early electron microscopy showed the kinetochore to have a trilaminar structure with a fibrous corona, apparent

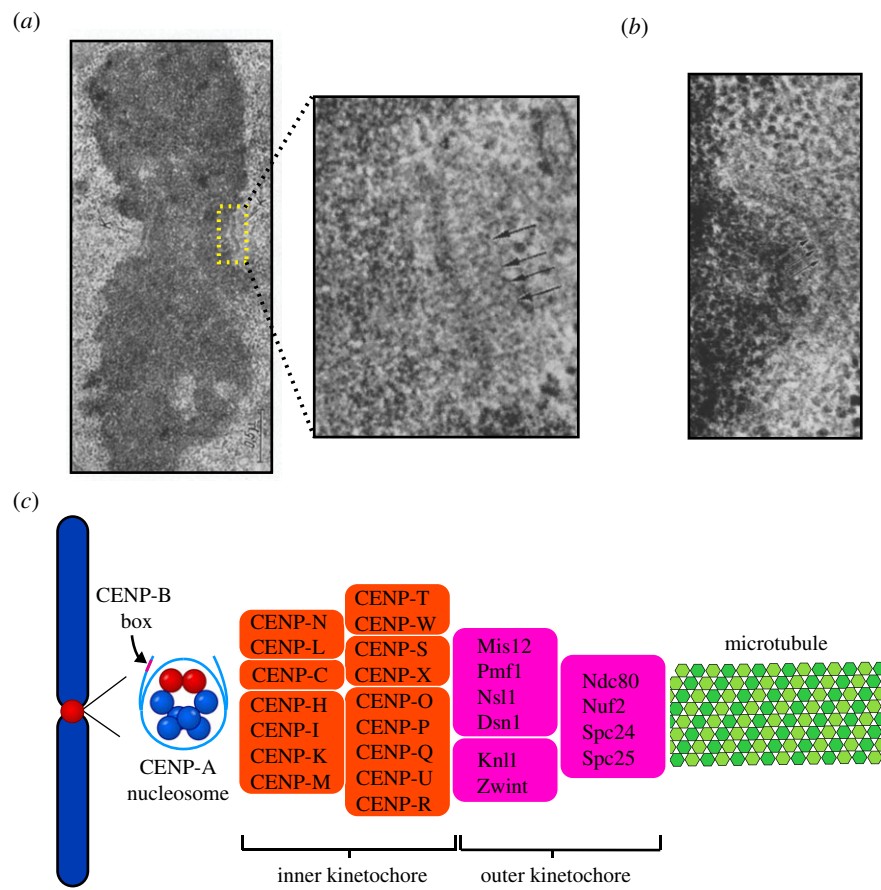

**Figure 1.** Structural organization of the centromere and kinetochore. (*a*) Electron micrograph of a mitotic chromosome with paired kinetochores on either side of the primary constriction (centromere), prior to microtubule attachments [3]. The inset shows a higher-magnification electron micrograph, which further reveals the trilaminar structure of the kinetochore, showing fibrous elements on either side of the dense central kinetochore plate. Arrows indicate kinetochore fibrils extending out from centromere. Permission to reproduce was sought from the author, Dr William Brinkley, but no response was received. (*b*) This micrograph of the centromere–kinetochore region shows a crescent-shaped kinetochore and the fibrous corona extending outwards from it. (*c*) A schematic of the DNA-microtubule interface, from the CENP-A nucleosomes found in centromeric DNA, through the many subunits of the kinetochore complex, and finally to the microtubule.

only when not bound to microtubules, that expands into a crescent shape prior to microtubule binding (figure 1*a,b*) [3,10,11]. There has been some debate as to whether this trilaminar structure was due to fixation techniques, and other imaging approaches find the kinetochore to be a fibrous mesh [11–13]. There is broad agreement, though, that the kinetochore provides attachment sites for microtubules that are substantially separated in space from the DNA surface.

In human cells, the foundation of the kinetochore is centromeric chromatin, which contains unusual nucleosomes where CENP-A takes the place of histone H3 [6,14,15]. CENP-B is a DNA-binding protein that interacts with centromeric DNA sequences [16–18]. Moving out from chromatin towards the microtubules, the next part of the kinetochore assembly is the inner kinetochore, which contains many CENP proteins organized in various subcomplexes including CENP-C, CENP-LN, CENP-HIKM, CENP-OPQUR and CENP-TWSX, together termed the constitutive centromere associated network (CCAN) (figure 1*c*) [19–31]. Building on top of the inner kinetochore is the outer kinetochore, containing the Ndc80, Knl1 and Mis12 complexes, which mediate connections to spindle microtubules (figure 1*c*) [31–42]. Our review addresses structural and biophysical features of the CENP-A nucleosome, the 16-subunit CCAN, and the manner in which these complexes form the basis for microtubule binding via the outer kinetochore and other microtubule-associated proteins. Many other proteins that help direct chromosome

segregation, including the inner centromeric chromosome passenger complex (CPC), components involved in sister chromatid cohesion and the components of the spindle assembly checkpoint (SAC), will not be discussed here but have been reviewed at length elsewhere [2,32,43–46]. Rather than comprehensively cover all kinetochore components, we outline the central elements of the centromere-to-microtubule connection. Recent advances in structural studies of the kinetochore have given us higher-resolution structures of individual kinetochore components. They provide new details about how they interact with each other, with centromeric DNA and with spindle microtubules. Here, we review this exciting progress and emphasize areas where the findings can be synthesized into emerging models for the mechanisms underlying chromosome segregation.

## 2. Definition of the centromere

Considering the fundamental nature of chromosome segregation, the chromosomal attachment site for binding microtubules exhibits surprisingly large variation across eukaryotes. In many heavily studied systems (budding yeast, fission yeast, fruit fly, mouse, human and others), chromosomes are monocentric, with a single centromere locus where the kinetochore forms and microtubules bind. Many others (worms, some insects, some plants and others) have holocentric

royalsocietypublishing.org/journal/rsob Open Biol. 10: 200051

chromosomes in which microtubules attach along the full length of the chromosome [47]. Our focus will be on monocentric chromosomes. In many species, centromeres are made up of repetitive DNA sequences [48–53]. The very small, so-called 'point' centromeres in the budding yeast *S. cerevisiae* are defined by a specific 125 bp centromere sequence [49,50]. However, in diverse eukaryotes, the centromere is defined epigenetically by the presence of nucleosomes containing CENP-A in place of conventional histone H3 [54]. In many cases, including humans, the location of the centromere is coincident with large stretches of highly repetitive DNA, where the smallest repeating unit is roughly the size of a nucleosome. In humans, for instance, the repetitive DNA is called α-satellite and the smallest repeating unit is 171 bp [51–53].

The human regional centromere is made up of a core of homogeneous ordered repeats and CENP-A nucleosomes flanked by outer regions of heterochromatin containing less organized repeats [55]. α-satellite repeats are organized in a higher-order repeat pattern [6,51–53]. α-satellite repeats are not required for specifying the centromere location [56]. They may participate in the formation and stability of pericentromeric heterochromatin, which forms the outer boundary of the centromere [57]. The CENP-B box (figure 2a) is a 17 bp motif that exists in a subset of α-satellite monomer repeats. CENP-B binds to this sequence (figure 2b; table 1) and contributes to centromere function [9,17,58–60]. The CENP-B box and CENP-B protein are absent from neocentromeres as well as from normal Y chromosomes [17,56]. Furthermore, deletion of CENP-B does not affect viability in mice [61–63]. However, the human Y chromosome, which lacks CENP-B, mis-segregates at a higher rate than other chromosomes [59]. The deletion of CENP-B also magnifies experimental insults to other centromere components [59]. CENP-B has a role in centromere establishment, since first-generation human artificial chromosomes (HACs) require CENP-B protein and a high density of CENP-B boxes on the α-satellite DNA HAC templates [55,64,65]. However, this function of α-satellite DNA and CENP-B can be completely bypassed when CENP-A nucleosomes are initially assembled on the HAC template upon its introduction into cells [66].

Beyond the presence of repetitive sequences at the centromere, there is also evidence that transcription of centromeric DNA, and the transcripts themselves, are involved in the loading of CENP-A nucleosomes at the centromere and the stabilization of kinetochore components [67–69]. CENP-A also carries many unique post-translational modifications that are probably involved in the epigenetic definition of the centromere. These post-translational modifications have been implicated in many processes central to the centromere including CENP-A incorporation and recruitment of inner kinetochore proteins (reviewed in [70]).

The centromere forms the foundation on which the kinetochore is built. It is the foundation for correct attachments of chromosomes to spindle microtubules and their proper segregation during mitosis or meiosis. For these reasons, the specification of the centromere and its propagation through generations is essential for genetic inheritance, and certainly for cell and organismal viability as well.

## 3. CENP-A nucleosomes

CENP-A nucleosomes form the foundation of the kinetochore assembly and are required, either directly or indirectly, for the localization of all known kinetochore components [71,72]. CENP-A is a variant of histone H3 and exists in chromatin as part of nucleosomes with conventional histones H2A, H2B and H4 [14,15,73,74]. CENP-A nucleosomes are interspersed with canonical H3 nucleosomes at the centromere, and the chromatin must fold in such a way as to expose CENP-A for kinetochore assembly [13,75]. Multiple models have been put forward to explain how this folding occurs in order to expose CENP-A nucleosomes on the surface of the mitotic chromosome [13,76] (reviewed in [6]).

The histone fold domain of CENP-A shares 62% sequence identity with that of histone H3, but its N-terminal 'histone tail' shares a basic character but no sequence identity with that of H3 (a large number of Lys residues in H3 are replaced with a large number of Arg residues in CENP-A) [74]. Within the histone fold, the CENP-A targeting domain (CATD) is composed of the first loop (L1) and second αhelix (α2), and is both necessary and sufficient for CENP-A targeting to the centromere (figure 2c) [77–80]. The CATD also houses the recognition site for the CCAN component CENP-N [81,82]. CENP-C, another CCAN component, recognizes a 6 a.a. sequence at the extreme C-terminal region of CENP-A [81]. The CATD also appears to support CENP-C recognition in *Xenopus* extracts and in early steps of mammalian centromere formation [83,84], although structural studies [85–87] would suggest this is likely to be an indirect effect through other physical properties of the CENP-A nucleosome imparted by the CATD [88].

CENP-A is incorporated into a nucleosome along with canonical histones H4, H2A and H2B with a histone stoichiometry similar to canonical nucleosomes (figure 2d) [88–91] (note that this was controversial, at least in some species [88,90,92–95], but the preponderance of data suggests that the major form in mammals—and probably in many other eukaryotes—is an octamer; reviewed in [54]), but the CENP-A nucleosome has some important physical differences. Arrays of CENP-A nucleosomes more readily form compact structures than their counterparts containing conventional histone H3 [96]. On the individual nucleosome level, however, they have looser connections with the DNA at the nucleosome 'entry and exit' sites relative to canonical nucleosomes, made even looser when CENP-C is engaged (figure 2e,f) [88,89,96–98]. In addition to differences in terminal DNA paths, CENP-A nucleosomes assembled on natural α-satellite DNA also have a distinctive superhelical bulge near superhelical location -3 to -5 (figure 2g) [86,91,99]. The α-satellite DNA sequence directs local and global changes in the structure of CENP-A nucleosomes that provide a foundation for CCAN assembly. Within the CENP-A nucleosome, CENP-A forms a more rigid contact point with histone H4 relative to the same point between H3 and H4 in canonical nucleosomes [79,80,88]. CENP-A also directs a steady-state nucleosome conformation in solution where the CENP-A/CENP-A' interface is rotated relative to the corresponding H3/H3' interface in canonical nucleosomes (figure 2h–j) [86–88,91,97,99,100]. Future work is needed to investigate how these features impact the assembly of higher-order complexes at the centromere (i.e. bound to the mammalian CCAN and with natural centromere DNA; more on this below).

The location of centromeres on chromosomes must be conserved across cell generations, so there must exist some mechanism to ensure that CENP-A is maintained at centromeric chromatin during mitosis. Indeed, the level of CENP-A at centromeres is stable across numerous cell divisions

royalsocietypublishing.org/journal/rsob    Open Biol. 10: 200051

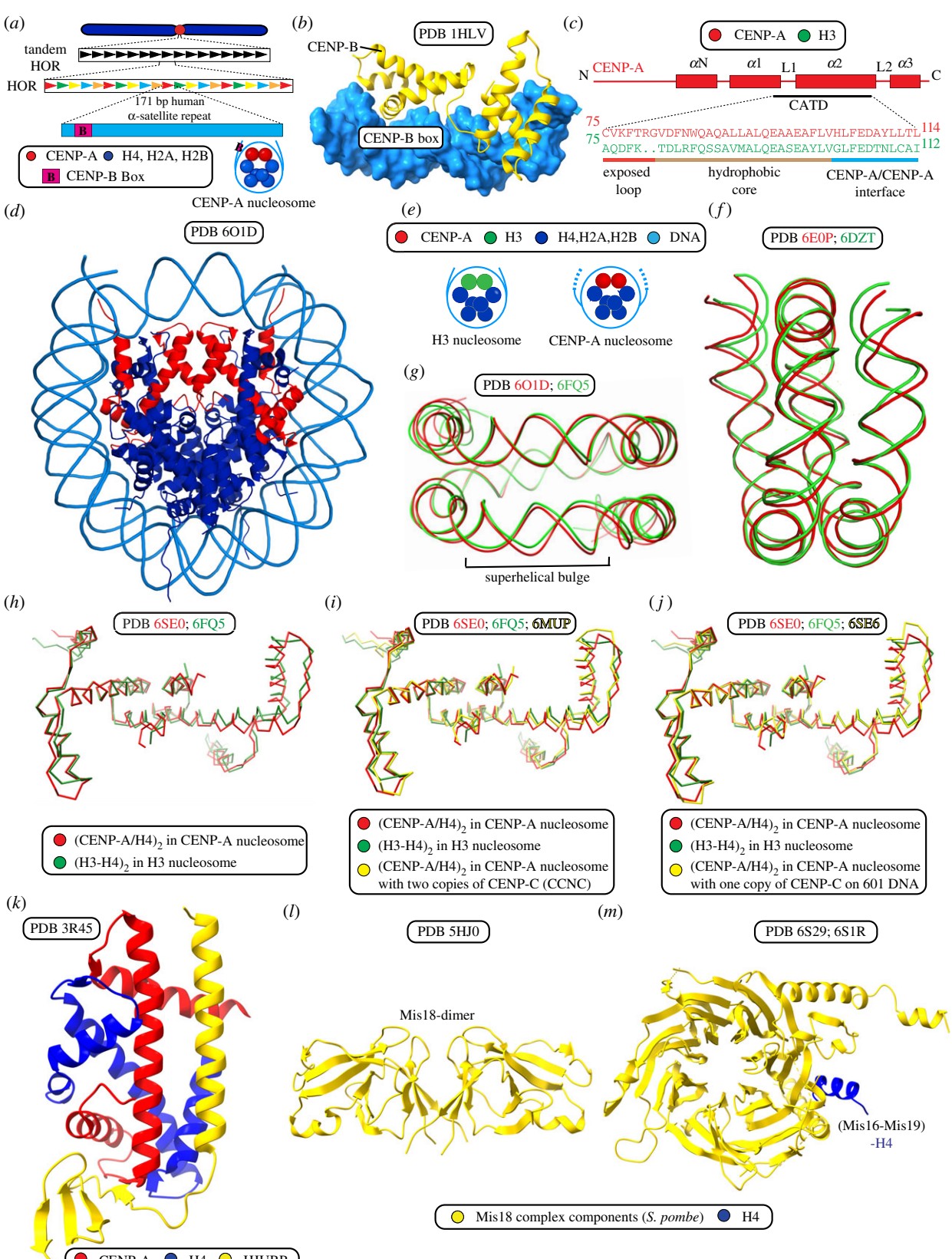

**Figure 2.** (*Caption overleaf.*)

[101,102]. CENP-A is partitioned between sister chromatids when DNA is replicated in S-phase [101,103], meaning that CENP-A levels at the centromere are counter-intuitively half-maximal at the crucial cell cycle stage when kinetochores are formed in mitosis. New incorporation of CENP-A happens in G1, after mitotic exit, whereas new H3 is incorporated during DNA replication [101,104,105]. This means that nascent CENP-A (in complex with its binding partner histone H4)

must displace H3/H4 when new CENP-A nucleosomes are incorporated in G1 [105].

HJURP is a histone chaperone specific for CENP-A/H4 that is critical for nascent CENP-A nucleosome incorporation in G1 [106,107] and for re-incorporation behind the replication fork [108]. HJURP localizes to the centromere only during G1, and its CENP-A binding domain specifically recognizes and binds to the CATD [106,107,109]. HJURP is also sufficient for

**Figure 2.** (*Overleaf.*) Structure and assembly of CENP-A chromatin. (*a*) Human centromeres typically are located within 0.5–5 Mb of α-satellite DNA arranged in large higher-order repeats (HOR) where the smallest repeating unit is 171 bp. The CENP-B box is located within the human 171 bp α-satellite centromeric repeat monomer, mostly outside of the CENP-A nucleosome DNA entry/exit site. (*b*) The structure of the human CENP-B N-terminal domain as bound to CENP-B box DNA. (*c*) Alignment of the CENP-A targeting domain (CATD) with the corresponding region of canonical histone H3. The CENP-A targeting domain (CATD) provides a distinct surface and features to centromeric CENP-A nucleosomes when compared with canonical H3 nucleosomes. The CATD provides three crucial characteristics: (1) CENP-A/CENP-A interface rotation; (2) strong interactions at the CENP-A/H4 interface due to hydrophobic stitch residues; (3) a protruding loop L1, which gives CCAN specificity) that combine to make the centromeric CENP-A nucleosome distinct. (*d*) Structure of a CENP-A nucleosome assembled on 145 bp human α-satellite DNA. (*e*) Schematic of DNA entry/exit dynamics. CENP-A nucleosomes show precise positioning on 171 bp α-satellite repeat and show flexibility in the terminal DNA predicted by access to MNase digestion. H3 nucleosomes lack these features when assembled on the same repeats. These features are important for the assembly of kinetochore proteins and are specific for CENP-A nucleosomes due to shortened α-1 helix in CENP-A. (*f*) DNA flexibility within the CENP-A nucleosome impacts the path of terminal DNA in all available cryo-EM and X-ray crystal CENP-A nucleosome structures relative to H3 nucleosome structures. The differences in the paths of DNA bound to CENP-A (red) and H3 (green) nucleosomes can be observed by alignment of the nucleosome cryo-EM structures using the (CENP-A/H4)$_2$ and (H3/H4)$_2$ dimers, as shown here. (*g*) A distinct feature of CENP-A nucleosome structures assembled on α-satellite DNA is the presence of a superhelical bulge, which is absent from H3 nucleosome structures and from CENP-A nucleosome structures assembled on 601 DNA or palindromic α-satellite DNA sequences. The superhelical bulge in the path of the nucleosomal DNA can be observed here by the alignment of CENP-A nucleosomes assembled on α-satellite DNA (red) with H3 nucleosomes assembled on 601 DNA (green) using the CENP-A/H4 dimer and H3/H4 dimer. The superhelical bulge of CENP-A nucleosomes provides a surface for accurate assembly of CCAN proteins. (*h–j*) Illustrations of the overall impact of intrinsic features of CENP-A and DNA sequence on nucleosome structure by alignment of the CENP-A/H4 dimer to the H3/H4 dimer or the CENP-A/H4 dimer, in the presence or absence of CENP-C from available structures. (*h*) CENP-A (red) and H3 (green) nucleosomes on 601 DNA. (*i*) CENP-A (red) and H3 (green) nucleosome on 601 DNA and CENP-A (yellow) nucleosome with CENP-C on α-satellite DNA. (*j*) CENP-A (red), H3 (green) nucleosome and CENP-A with CENP-C nucleosome (yellow) on 601 DNA. (*k*) The structure of the human CENP-A/H4 dimer in complex with its chaperone, HJURP. (*l*) Crystal structure of the CENP-A nucleosome assembly regulator, *S. pombe* Mis18. The structure includes its N-terminal Yippee-like domain, which is known to act as centromere targeting domain and contains a cradle-shaped pocket which binds DNA and is required for Mis18 functions. (*m*) Structure of the CENP-A nucleosome assembly regulatory complex member, *S. pombe* Mis16 with histone H4.

**Table 1.** Table of PDB structures. Colour coding to match the figures: yellow, CENP-A nucleosome assembly machinery; blue, structures related to the CENP-A nucleosome and its binding proteins; orange, CCAN components; magenta, outer kinetochore components.

| PDB | description | organism | figure | reference |
|------|------------|----------|--------|-----------|
| 1HLV | CENP-B bound to CENP-B box | *H. sapiens* | 2b | [60] |
| 6O1D | CENP-A nucleosome with native α-satellite DNA | *H. sapiens* | 2d,g, 3b | [91] |
| 6E0P | CENP-A nucleosome with native α-satellite DNA bound with antibody | *H. sapiens* | 2f | [91] |
| 6DZT | canonical nucleosome with native α-satellite DNA bound with antibody | *H. sapiens* | 2f | [91] |
| 6FQ5 | canonical nucleosome on 601 DNA | *H. sapiens* | 2g–j | [99] |
| 6SE0 | CENP-A nucleosome on 601 DNA | *H. sapiens* | 2h–j | [87] |
| 6MUP | CCNC on α-satellite DNA with two copies of CENP-C and CENP-N | *H. sapiens* | 2i, 3e, 4d | [86] |
| 6SE6 | CENP-A nucleosome on 601 DNA with one copy of CENP-C | *H. sapiens* | 2j, 3c | [87] |
| 3R45 | HJURP bound to H4/CENP-A heterodimer | *H. sapiens* | 2k | [112] |
| 5HJ0 | Mis18 complex | *S. pombe* | 2l | [124] |
| 6S29 | Mis16-Mis19 complex | *S. pombe* | 2m | [118] |
| 6S1R | Mis16-H4 complex | *S. pombe* | 2m | [118] |
| 6C0W | CENP-A nucleosome on 601 DNA with one copy of CENP-N | *H. sapiens* | 3d | [135] |
| 6MU0 | CCNC on α-satellite DNA with two copies of CENP-C and one copy CENP-N | *H. sapiens* | 3f | [86] |
| 6NUW | Ctf19 complex | *S. cerevisiae* | 4b,d, 5a–d | [145] |
| 6QLD | Ctf19 complex bound to CENP-A nucleosomes on 601 DNA | *S. cerevisiae* | 4c | [144] |
| 4P0T | CENP-M | *H. sapiens* | 4d, 5a–d | [147] |
| 3VH5 | CENP-T histone fold-WSX complex | *G. gallus* | 4d, 5a–d | [148] |
| 3VZA | crystal structure of Spc24 and Spc25 bound with CENP-T | *G. gallus* | 6b | [181] |
| 3IZ0 | Ndc80 complex 'bonsai' form bound to microtubule | *H. sapiens, B. taurus* | 6c | [182] |
| 5TCS | Ndc80 complex 'dwarf' form | *S. cerevisiae* | 6d | [183] |
| 5LSK | Mis12 bound to a fragment of CENP-C | *H. sapiens* | 6e | [193] |
| 4NF9 | Knl1/Nsl1 complex | *H. sapiens* | 6f | [198] |
| 4AJ5 | Ska complex | *H. sapiens* | 6g | [206] |

incorporation of CENP-A nucleosomes at an ectopic site such as in the formation of HACs [66,110,111]. A high-resolution structure of HJURP in complex with a CENP-A/H4 heterodimer (figure 2k) [112] revealed that the C-terminal β-sheet domain of HJURP occludes a major portion of the DNA binding surface of CENP-A/H4. This structure also revealed that, beyond the CATD, Ser68 on the α1-helix of CENP-A also contributes to the binding surface with HJURP [112]. HJURP is restricted from engaging with centromeres and depositing nascent CENP-A protein until cell cycle-dependent restrictions are relieved [113]. HJURP is directly recruited by the Mis18 complex composed of MIS18α, MIS18β and M18BP1 (also known as KNL2) (figure 2l) [114–118]. The fission yeast *S. pombe* lacks M18BP1, and in this species Mis18 instead forms an analogous complex with Mis16, Mis19 and Mis20 (figure 2m) [119–122]. M18BP1 also interacts with CENP-C, which may help to explain how it is targeted to centromeric chromatin and regulates CENP-A deposition [83,123]. A high-resolution structure from the fission yeast *S. pombe* revealed the structure of Mis18 in that species (figure 2l) [124]. Interestingly, in the fruit fly, *D. melanogaster*, the HJURP counterpart, Cal1, binds CENP-A by enveloping the CENP-A/H4 heterodimer [125]. Cal1 also guides the deposition of CENP-A at the centromere and directly binds CENP-C, thereby combining the roles of HJURP and the Mis18 complex [125–128]. Understanding exactly how these proteins and others target, deposit and maintain CENP-A at the centromere represents an important challenge for the field.

## 4. Recognition of CENP-A nucleosomes by CENP-C and CENP-N

Surface features on the core of the CENP-A nucleosome distinguish it from canonical nucleosomes containing conventional histone H3 so that it can be recognized by CCAN subunits CENP-C and CENP-N (figure 3) [91].

CENP-C is the 'keystone' of the inner kinetochore. It is an elongated protein with binding sites for several centromere and kinetochore components, serving to assemble and organize components of the CCAN [9,26,31,81,129,130]. In mammals, CENP-C has two nucleosome binding domains, the central domain and the CENP-C motif (figure 3a) [85]. The CENP-C motif is conserved from budding yeast, and in some species this is the only nucleosome-binding domain of CENP-C. The central domain is conserved only within mammals and has an especially high specificity for CENP-A over H3 compared with the less specific CENP-C motif [85], although recent findings with chicken CENP-C indicate that CDK-mediated phosphorylation of the CENP-C motif can enhance CENP-A nucleosome binding [131]. In humans, the central domain directs the structural transition in the CENP-A nucleosome that occurs upon CENP-C binding and stabilizes CENP-A nucleosomes [97,100,132], an especially important CENP-C activity during DNA replication [103]. In budding yeast, CENP-C^Mif2P interacts not only with the histone core of the CENP-A nucleosome but also with the especially AT-rich budding yeast centromere DNA through a continuous DNA and histone-binding surface [133].

The majority of CENP-C is thought to be elongated and largely disordered, so we lack a complete structural model. One globular domain is known to exist: the C-terminal portion of CENP-C, which contains a cupin domain that facilitates homodimerization (figure 3a). The CENP-C cupin domain consists of a nine-strand jelly roll folded architecture [134]. This central element of the CENP-C cupin domain is present in budding yeast as well as in organisms with regional centromeres, but the cupin domain of organisms with regional centromeres also contains additional structural features that stabilize its structure and contribute to homodimerization [134].

Along with CENP-C, CENP-N also directly interacts with CENP-A nucleosomes (figure 3c–f) [81,82]. Together, CENP-A, CENP-C and CENP-N form the core centromeric nucleosome complex (CCNC) [86,132]. Structures are available for CCNC components both alone and in various combinations (figure 3c–f) [86,87,91,134–137]. The N-terminal portion of CENP-N binds to the CATD of CENP-A (figure 3a) [82,132]. This interaction fastens CENP-A to nucleosomal DNA and stabilizes the CENP-A surface bulge [132], supporting the idea that CENP-N provides additional stability to centromeric chromatin beyond what can be attributed to the central domain of CENP-C. The binding of CENP-N to CENP-A is also stabilized by electrostatic interactions with nucleosomal DNA, an observation that is consistent with the coevolution of CENP-N and CENP-A [135,136]. The N-terminal domain of CENP-N does not interact directly with the central domain of CENP-C, but the N-terminal tail of H4 is sandwiched between the C-terminal end of the CENP-C central domain and two loops extending from the CENP-N N-terminal domain, providing a site of indirect interaction between the two non-histone components of the CCNC [86]. When the CCNC is assembled on its favoured natural DNA sequence (α-satellite DNA), there is a distinctive bulge four turns of DNA from the dyad axis that widens the superhelical pitch, making this structure an outlier compared with other nucleosomal structures (figure 2g) [86,89,99,135,138]. This DNA bulge probably accommodates a DNA shift favourable for the binding of the CENP-N N-terminal region, allowing two CENP-N molecules to interact equivalently with the nucleosome, one on each surface. Upon mitotic entry, CENP-C levels are constant per unit of centromeric chromatin, while CENP-N levels drop by half, leading to a model wherein the interphase form of the CCNC contains two copies each CENP-C and CENP-N, while the mitotic form contains two copies of CENP-C but only a single copy of CENP-N (figure 3e,f) [86].

## 5. The inner kinetochore: CCAN structure

Biochemical reconstitution and cell-based studies have revealed that, with the exception of CENP-C, the CCAN is composed of four constitutive subcomplexes, each of which is stabilized by reciprocal interactions among its component parts. These subcomplexes are CENP-LN, CENP-HIKM, CENP-TWSX and CENP-OPQUR (figure 4a) [22,23,26,139]. As discussed above, CENP-C and CENP-N have both been shown to directly interact with CENP-A and serve as important connections between the inner kinetochore and centromeric chromatin. Most human CCAN proteins have orthologues in the budding yeast (*S. cerevisiae*) Ctf19 complex, which also contains the CBF3 complex that directly binds yeast centromeric DNA and is not present in humans [25,37,140–143].

Major recent progress has been made in understanding the structure of the budding yeast CCAN (also referred to as Ctf19 complex) (figure 4b) [144–146]. These structures contain CENP-I^Ctf3, CENP-H^Mcm16, CENP-K^Mcm22, CENP-L^Iml3, CENP-N^Chl4, CENP-P^Ctf19, CENP-O^Mcm21, CENP-Q^Okp1, CENP-U^Ame1,

royalsocietypublishing.org/journal/rsob Open Biol. **10**: 200051

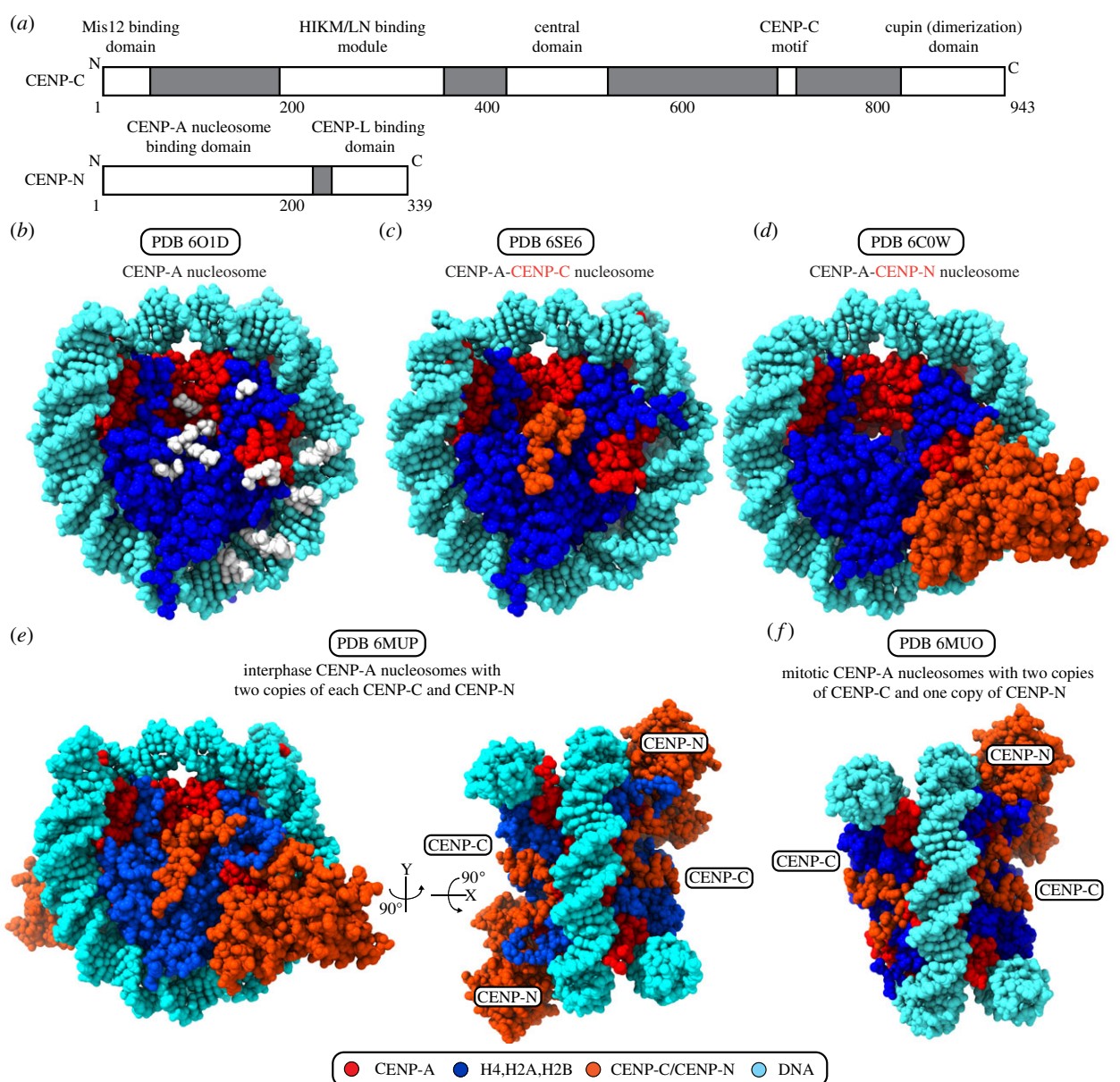

**Figure 3.** The CCNC forms the foundation of the kinetochore. (*a*) Domain coordinates for human CENP-C and CENP-N. (*b*) The CENP-A nucleosome structure is shown facing the surface of the histone octamer, with the dyad at top. White positions highlight the points on the protein and DNA surface where CCNC components have direct contacts. (*c*) The CENP-A nucleosome assembled with Widom 601 DNA and bound with the central domain of CENP-C. (*d*) The CENP-A nucleosomes assembled with Widom 601 DNA and bound with CENP-N. (*e*) The CENP-A nucleosome assembled with human α-satellite DNA and bound with two copies of CENP-C and CENP-N, proposed to be the interphase form of the CCNC. (*f*) The CENP-A nucleosome assembled with human α-satellite DNA and bound with two copies of CENP-C and one copy of CENP-N, proposed to be the mitotic form of the CCNC.

CENP-T[Cnn1] and CENP-W[Wip1], as well as Nkp1 and Nkp2, which apparently lack human homologues. Other crystallographic studies have reported the structures of human CENP-M [147] and chicken CENP-TWSX [148]. One of the recent larger structures is of the budding yeast CCAN bound to CENP-A nucleosomes and also includes CENP-C (although most of CENP-C did not produce assignable density except for the small region contacting CENP-A nucleosomes). CENP-S[Mhf1] and CENP-X[Mhf2] orthologues were not included, and there are not apparent CENP-M or CENP-R orthologues present in budding yeast [144]. In this structure (figure 4*c*), the orientation of CENP-N places it nowhere near the contact point on CENP-A or the proximal DNA (at SHL 2.5–3.5) described for mammalian CENP-N orthologues (figure 3). Instead, it contacts nucleosomal DNA in an opposite location (SHL 6), indicating that budding yeast CENP-N may not share the direct recognition of CENP-A protein possessed by

its mammalian counterparts. The nucleosome in this structure [144] used the strong artificial nucleosome positioning sequence, Widom 601 [149]. While two copies of CENP-C bind to budding yeast CENP-A nucleosomes when assembled on natural centromere DNA [133,144], only a single CENP-C binds when the Widom 601 sequence is used [144]. A parallel situation is present in mammals, where nucleosomes assembled with the natural centromere DNA (human α-satellite) more easily assemble into the CCNC than do those formed on the artificial Widom 601 sequence [86]. This is probably due to the natural flexibility of α-satellite DNA at key locations along the nucleosome surface where CENP-C and CENP-N bind (figure 4*d*). These findings drive home the importance of keeping the choice of DNA sequence in mind when one considers the proposed structural models, since it is possible that artificial sequences may not report accurately on the stoichiometry or local arrangement of CCAN components.

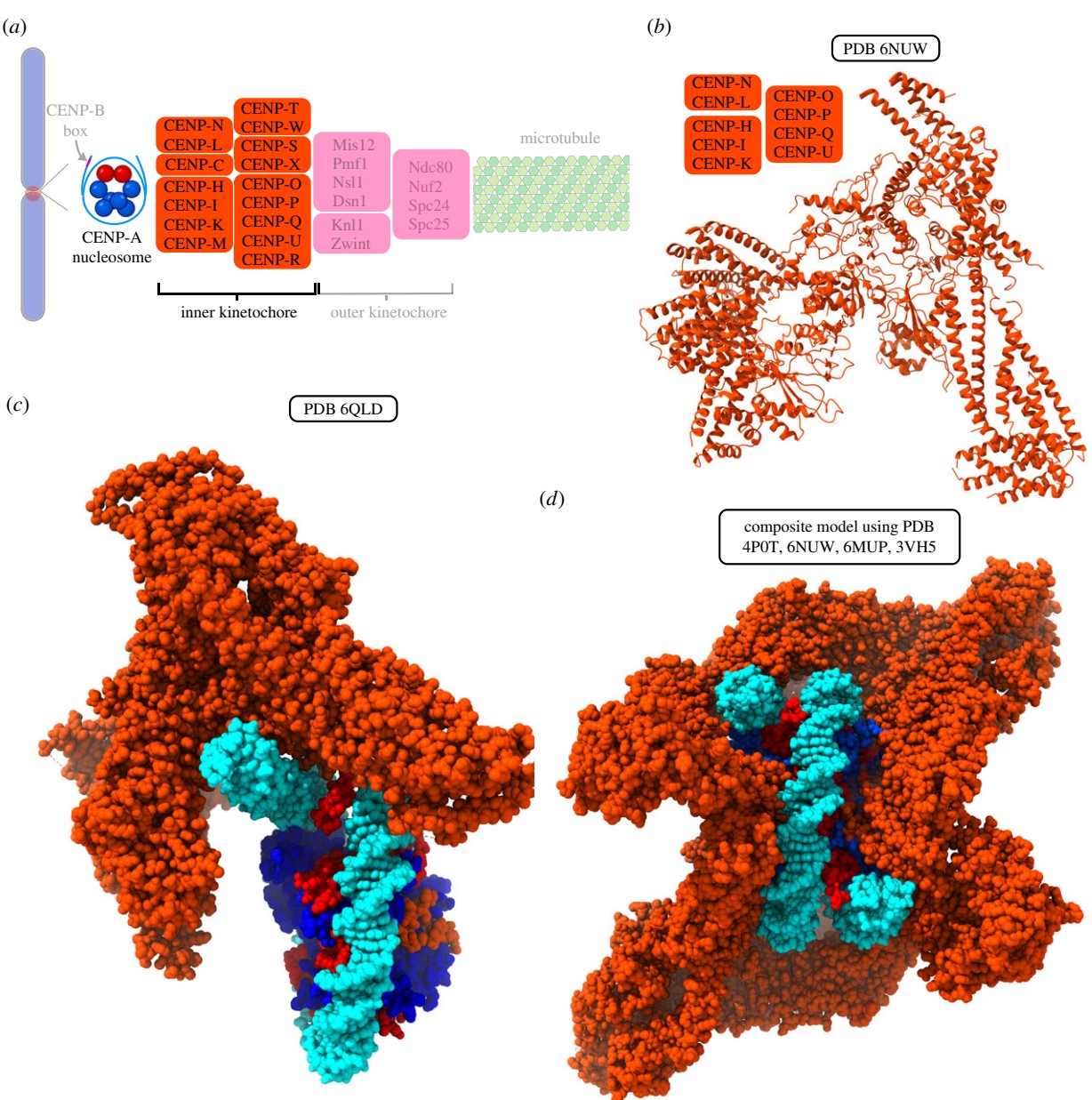

**Figure 4.** Structure of the yeast CCAN and models of the human CCAN complex bound to CENP-A nucleosome. (*a*) A schematic of the DNA–microtubule interface, with the CENP-A nucleosome and inner kinetochore highlighted. (*b*) Structure of the *S. cerevisiae* Ctf19/CCAN complex containing homologues to human CENP-LN, CENP-HIK and CENP-OPQUR. (*c*) Cryo-EM structure of one copy of *S. cerevisiae* Ctf19 (lacking homologues of CENP-M and CENP-TWSX) interacting with CENP-A nucleosome on Widom 601 DNA. In this structure, CENP-N$^{Chl4}$ does not contact CENP-A or proximal DNA at the sites described in structures of mammalian CENP-N. (*d*) Composite model of CCAN (Ctf19 complex/CENP-M/CENP-TWSX) with the CCNC. Structural alignment was performed by aligning the N-terminal domain of the common subunit, CENP-N.

# 6. CENP-C and CENP-LN

Both CENP-C and CENP-N are required to recruit all other CCAN components [26,81–83,85,130,135,136,150]. Thus, when asymmetry arises on an individual CENP-A nucleosome in CENP-C or CENP-N copy number (i.e. in the proposed asymmetric mitotic form of the CCNC) [86], it follows that the surface of the CENP-A nucleosome bound by both CENP-C and CENP-N would recruit the full CCAN capable of nucleating kinetochore formation.

One component, CENP-C, acts as a scaffold with modular binding surfaces that can interact with diverse components (CENP-A nucleosomes, CENP-A assembly proteins, CCAN components, self-interactions and outer kinetochore components [26,81,130,132,150,151]), spanning a distance of

approximately 100 nm (figure 3*a*) [152]. In contrast, the other component, CENP-N, is only known to interact with CENP-A nucleosomes via its N-terminal globular domain and with its closest binding partner within the CCAN, CENP-L, via its C-terminal domain (figure 3*a*) [82,132,135,136]. Less is known about CENP-L, itself. As a whole, the CENP-LN subcomplex interacts preferentially with CENP-A over H3 [31]. CENP-L$^{Iml3}$-CENP-N$^{Chl4}$ from budding yeast, in the context of the CCAN/Ctf19 complex (figure 5*a*), is thought to be similar in structure to human CENP-LN [145]. The budding yeast structure revealed that the CENP-N$^{Chl4}$ linker domain (which links the N-terminal and C-terminal regions) makes extensive contacts with other CCAN/Ctf19 components, CENP-P$^{Ctf19}$-CENP-O$^{Mcm21}$ [145]. In addition, the β3-β4 loop of CENP-N$^{Chl4}$ extends into the central cavity of the CCAN/Ctf19

royalsocietypublishing.org/journal/rsob Open Biol. **10**: 200051

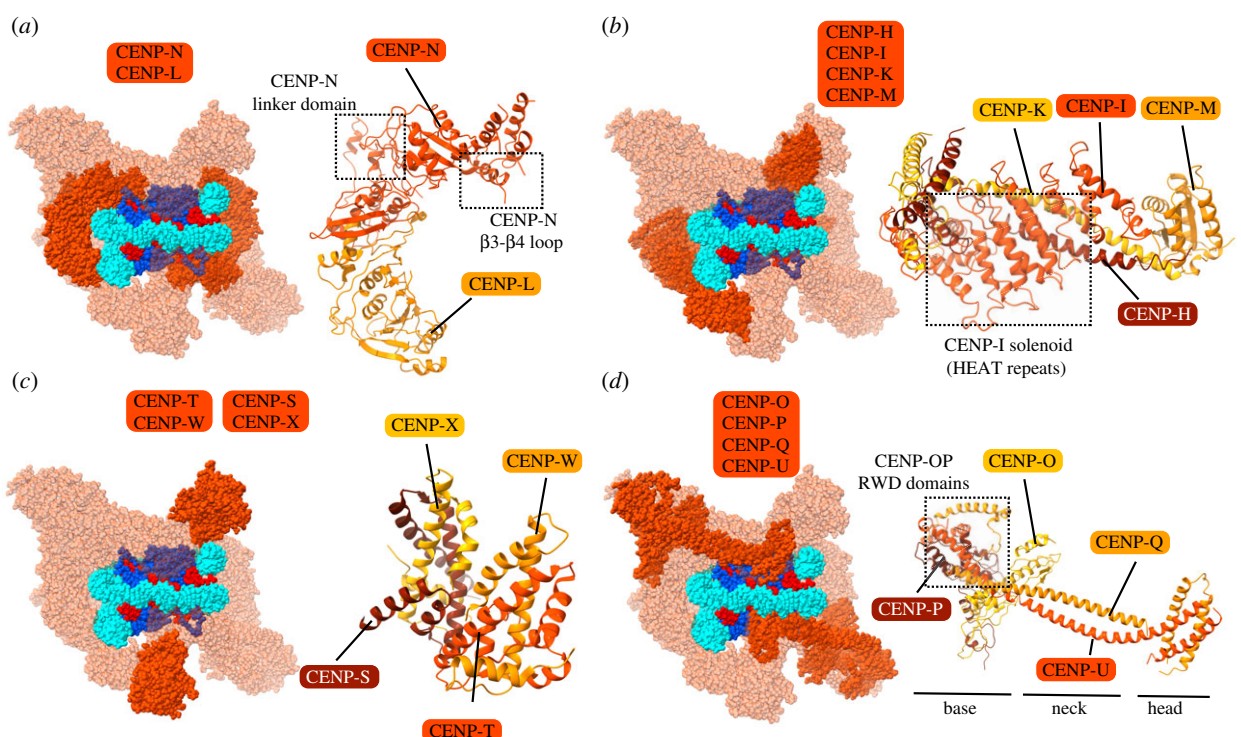

**Figure 5.** Organization of CCAN sub-complexes on CENP-A nucleosomes. (*a*–*d*) Location of indicated complex within the composite model of CCAN-CCNC complex (left) and ribbon diagram in isolation (right).

complex and contacts the coiled coils of CENP-U^Ame1^-CENP-Q^Okp1^ [145]. Finally, CENP-L^Iml3^ binds directly to the Ctf3 complex (CENP-HIK) providing a link bridging to other components of the CCAN/Ctf19 to the CENP-A nucleosome [145].

# 7. CENP-HIKM

CENP-HIKM is thought to be a single tetrameric unit that is recruited to the kinetochore by CENP-C, and it can bind to CENP-C and CENP-LN within a larger complex: CENP-CHIKMLN [139,145,147]. CENP-HIKM contributes to the CENP-A nucleosome binding affinity of CCNC components, but it alone is not selective for CENP-A nucleosomes relative to canonical nucleosomes containing H3 [139]. CENP-M is a pseudo-GTPase that is required to stabilize CENP-I and is essential to the stability of the CENP-HIKM complex as a whole [147]. Disruption of the interaction between CENP-M and CENP-I leads to defective kinetochore assembly and chromosome alignment [147].

A structure of the budding yeast Ctf3 complex (CENP-HIK) in the context of the CCAN/Ctf19 complex reveals the organization of this subcomplex (figure 5*b*). The C-terminal region of CENP-I^Ctf3^ has multiple HEAT repeats, and a CENP-H^Mcm16^-CENP-K^Mcm22^ coiled-coil motif fits into the concave surface of the CENP-I^Ctf3^ HEAT array [146]. The N-terminal region of CENP-I^Ctf3^ was separately shown to contain HEAT repeats just like the C-terminal region, and the interaction between the C-terminal region of CENP-I^Ctf3^ and CENP-H^Mcm16^-CENP-K^Mcm22^ is mirrored at the other end of the complex [153]. The CENP-H^Mcm16^-CENP-K^Mcm22^ dimer extends through the entirety of this structure. A tight space between the CENP-I^Ctf3^ HEAT array and the CENP-H^Mcm16^-CENP-K^Mcm22^ helices provides a binding site for another CCAN/Ctf19 component, CENP-O^Mcm21^ [145,146]. CENP-M lacks a

homologue in yeast and so is absent in the CCAN/Ctf19 complex [144–146]. However, a high-resolution structure of CENP-M in isolation is available and is included in the composite CCAN model (figure 5*b*) [147]. In the CENP-HIKM complex, CENP-M has been proposed to bind within the concave surface of the CENP-I solenoid [147].

# 8. CENP-TWSX

The tetrameric CENP-TWSX complex is made up of the 2-unit CENP-TW and CENP-SX subcomplexes (figure 5*c*) [23,148]. Each component of the CENP-TWSX complex has a histone fold domain [148,154–156]. Unlike CENP-A, CENP-TW turns over frequently throughout the cell cycle, and is incorporated at the centromere in late S and G2 [157]. CENP-T recruitment to the kinetochore involves the CENP-HIKM complex and the N-terminal tail of CENP-A [84,158,159]. Ablating either CENP-TW or CENP-SX leads to the destabilization of the outer kinetochore [22,23]. While in vertebrates CENP-T and CENP-W are both essential proteins, CENP-S and CENP-X are not [22,23]. Nonetheless, in one proposal CENP-TWSX forms a tetrameric structure that organizes DNA in a nucleosome-like complex [148,160]. DNA binding by the CENP-TWSX complex relies on the histone folds of CENP-T and CENP-W and does not appear to exhibit any sequence specificity [148,160]. The CENP-TWSX complex induces positive DNA supercoiling, opposite to the negative supercoiling generated upon canonical nucleosome assembly [148,160]. In the CCAN/Ctf19 complex, CENP-T^Cnn1^ contacts CENP-I^Ctf3^ through an extension of its histone fold motif composed of two additional short helices [145,161]. Furthermore, the orientation of CENP-T^Cnn1^-CENP-W^Wip1^ in the overall CCAN/Ctf19 complex is consistent with binding to DNA [145].

## 9. CENP-OPQUR

The 5-subunit CENP-OPQUR complex (CENP-U is also known as CENP-50) is an important connection between the inner and outer kinetochore. CENP-OPQUR is recruited by the CENP-CHIKMLN complex and binds to a joint interface of CENP-HIKM and CENP-LN [31]. CENP-OP may form a bridge between CENP-CHIKMLN and CENP-QUR. A recent 26-subunit reconstruction of human kinetochore components revealed that CENP-OP and CENP-QU are constitutive subcomplexes whereas CENP-R is stable and soluble in isolation [31]. Similarly, kinetochore recruitment of the units making up the CENP-OPQU complex are interdependent, but CENP-OPQU can localize to the kinetochore in the absence of CENP-R. CENP-R interacts primarily with the CENP-QU subcomplex and does not appear to be required for recovery from spindle damage, unlike the other components of the CENP-OPQUR complex [31]. Although this reconstruction did not yield high-resolution structural information, it showed the CENP-OPQU complex to be bi-lobed with a smaller head and larger base, with the addition of CENP-R adding a protuberance to the neck of the complex and enlargement of its base [31].

Budding yeast do not have a homologue for CENP-R, but the yeast COMA complex is equivalent to human CENP-OPQU (figure 5d). The COMA complex is made up of CENP-P$^{Ctf19}$, CENP-Q$^{Okp1}$, CENP-O$^{Mcm21}$ and CENP-U$^{Ame1}$ [37,141]. CENP-U$^{Ame1}$ has been shown to interact with the outer kinetochore MIND complex, analogous to the human Mis12 complex [150], discussed below. The recent high-resolution structures of the yeast CCAN/Ctf19 complex provides more structural detail about this complex [144–146]. CENP-Q$^{Okp1}$ and CENP-U$^{Ame1}$ are structurally intertwined with Nkp1 and Nkp2, which have no clear mammalian orthologues. CENP-Q$^{Okp1}$-CENP-U$^{Ame1}$ and Nkp1/Nkp2 both have helical hairpins in their N-terminal regions, which interact to form a bundle of four helices. The C-terminal regions of the same four proteins form a four-chain helical coil [145,150,162]. CENP-O$^{Mcm21}$ and CENP-P$^{Ctf19}$ have C-terminal RWD domains that interact with the intermediate segments of CENP-Q$^{Okp1}$, CENP-U$^{Ame1}$ and Nkp1 [145,162,163]. CENP-O$^{Mcm21}$ and CENP-P$^{Ctf19}$ were also shown to have N-terminal extensions that are flexible in isolation [145,162]. In the CCAN/Ctf19 complex structure, a helix at the tip of CENP-O$^{Mcm21}$ was shown to interact with the C-terminal region of the CENP-I$^{Ctf3}$ solenoid [145]. Furthermore, deletion of this helix caused CENP-I$^{Ctf3}$ to fail to localize to the kinetochore, suggesting that CENP-O$^{Mcm21}$ may be involved in recruiting the CENP-HIK$^{Ctf3}$ complex to the kinetochore [145]. Nkp1 and Nkp2, crucial elements of this structure in budding yeast, are absent in mammals, as mentioned above, making it unclear to what degree some of the structural information can be extrapolated to the human kinetochore.

The precise role of CENP-OPQUR at the kinetochore remains unclear, but it has been reported to be required for recruiting to the mitotic kinetochore the motor protein, CENP-E, and the mitotic regulatory kinase, PLK1 [164–166]. CENP-QU can bind to microtubules and is capable of sustaining long-term attachments to them *in vitro* [31,167,168]. Relevant to this activity is that the structure of the N-terminal region of CENP-Q is similar to that of the microtubule-binding outer kinetochore protein Ndc80 [31].

## 10. The outer kinetochore

The outer kinetochore builds on top of the inner kinetochore and is the site of microtubule binding as well as the site of SAC recruitment [7]. The outer kinetochore transduces force from depolymerizing microtubules to ensure proper segregation of sister chromatids towards the spindle poles. The outer kinetochore is primarily made up of the 10-subunit KMN assembly, which contains the Ndc80 complex, Mis12 complex and the Knl1 complex (figure 6a) [33,35,39,42,169–171]. There are also numerous proteins beyond the KMN assembly that localize to the outer kinetochore [33,37]. In vertebrates, the complexes of the KMN assembly are not localized to the kinetochore in G1 but instead are recruited in S and G2 phases, with the Ndc80 complex the last to be recruited [172–175]. Thus, the process of preparing the centromere for mitosis is thought of as a stepwise recruitment of outer kinetochore components.

## 11. Ndc80 complex

The four-subunit Ndc80 complex is made up of Ndc80, Nuf2, Spc24 and Spc25, and functions as the primary microtubule receptor on the kinetochore [171,176,177]. The complex is composed of two heterodimers: Ndc80-Nuf2 and Spc24-Spc25 [42,178–180]. The Ndc80-Nuf2 heterodimer has a globular N-terminal region that mediates microtubule binding as well as a C-terminal coiled coil [178–180]. Similarly, the Spc24-Spc25 heterodimer has a C-terminal globular region containing paired RWD domains as well as an N-terminal coiled coil [178–180]. The RWD domains of Spc24 and Spc25 interact with the disordered N-terminal region of CENP-T, allowing up to two Ndc80 complexes to be recruited for each copy of CENP-T (figure 6b) [161,174,181]. Each pair of proteins interacts via their coiled-coil domains for dimerization, and then the two heterodimers interact through their dimerized coiled-coil domains to form a tetramer [178–180]. This gives the Ndc80 complex an overall dumbbell-like shape, with a globular domain at each end and coiled coils in between [178–180]. Due to the elongated shape of the Ndc80 complex, no high-resolution structures exist of the full complex in its native form. However, two modified forms provide important structural information that can be extrapolated to understand the structure of the complex as a whole. One such modified structure is the 'bonsai' structure, a chimeric Ndc80 complex containing minimal coiled-coil domains, in which Ndc80 is fused to Spc25 and Nuf2 is similarly fused to Spc24 (figure 6c) [180,182]. The other is the 'dwarf' form, which is shortened with respect to the full complex but maintains the full tetramer junction (figure 6d) [183].

The Ndc80 complex binds microtubules via N-terminal calponin homology domains in Ndc80 and Nuf2 (figure 6c) [182,184,185]. The basic N-terminal domain of Ndc80 interacts with the acidic E-hook on both α- and β-tubulin [184–186]. Unlike Ndc80, Nuf2 does not contact microtubules directly through its calponin homology domain, but the domain is necessary for high-affinity binding of the Ndc80 complex to microtubules [180,187]. The N-terminal domain of Ndc80 has also been implicated in cooperative binding with other Ndc80 complexes bound along the length of the same protofilament [182,184]. The oligomerization of the Ndc80 complex through the Ndc80 subunit means that a single Ndc80 complex can detach from the microtubule where tubulin is

royalsocietypublishing.org/journal/rsob   Open Biol. 10: 200051

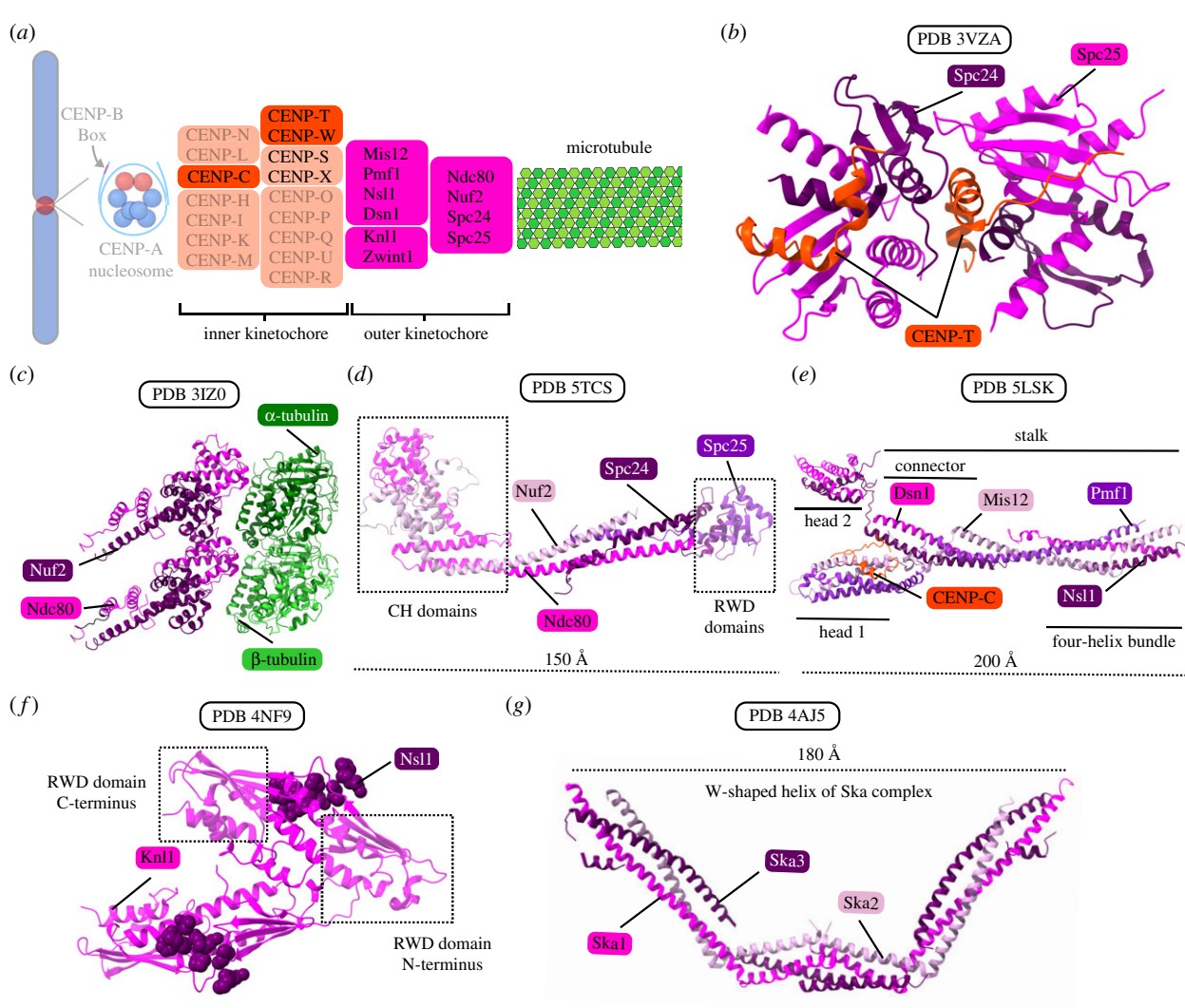

**Figure 6.** The KMN complex and other microtubule couplers. (*a*) A schematic of the DNA-microtubule interface, highlighting the microtubule, outer kinetochore and components of the inner kinetochore that couple the KMN to CENP-A nucleosomes. (*b*) Crystal structure of chicken Spc24/Spc25 globular domain (part of the Ndc80 complex) bound with CENP-T. This interaction allows up to two Ndc80 complexes to be recruited for each copy of CENP-T. (*c*) Structure of the *S. cerevisiae* 'bonsai' chimeric Ndc80 complex bound to tubulin. The 'bonsai' structure contains minimal coiled-coil domains, and Ndc80 is fused to Spc25 and Nuf2 to Spc24. The Ndc80 complex binds microtubules via N-terminal calponin homology domains in Ndc80 and Nuf2. (*d*) Structure of the dwarf Ndc80 tetramer, which is shortened with respect to the full complex but maintains the full tetramer junction. (*e*) Crystal structure of the human Mis12 complex bound with a fragment of the CENP-C N-terminal region. The Mis12 complex also interacts with CENP-T, Knl1 and Ndc80, serving as an important interaction hub between the KMN assembly and the inner kinetochore. (*f*) Crystal structure of the C-terminal RWD domains of human Knl1 interacting with Nsl1 residues. Knl1 is the largest structural subunit in the outer kinetochore and provides binding sites for Nsl1 as well as several proteins that interact with the outer kinetochore. (*g*) Crystal structure of the human Ska complex. The Ska complex functions to enhance the microtubule binding ability of the Ndc80 complex.

depolymerizing but remain associated with the microtubule through oligomerization [182,184]. This mechanism probably contributes to how the kinetochore can maintain load-bearing attachments to depolymerizing microtubules. It has been shown that, unlike individual Ndc80 complexes in solution, Ndc80 complexes immobilized at high concentration on beads can create load-bearing attachments to depolymerizing microtubules [188–190]. This evidence underscores the importance of cooperative binding in forming stable microtubule attachments. Finally, the Ndc80 complex, and the N-terminal domain of Ndc80 in particular, may also act as a conformation sensor for straight protofilaments [182,191].

Aurora B kinase, a key regulator of kinetochore–microtubule attachments and the catalytic subunit of the CPC, phosphorylates sites on the N-terminal tail of Ndc80, neutralizing its positive charge and thereby decreasing its microtubule binding affinity [182,184]. In this way, phosphorylation by Aurora B kinase serves as a negative regulator of kinetochore–

microtubule attachments at the Ndc80 complex. The related Aurora A kinase, which is involved in spindle pole separation in early mitosis, has also been shown to regulate kinetochore–microtubule dynamics in metaphase via phosphorylation of a site in the N-terminal tail of Ndc80 [177].

## 12. Mis12 complex

The four-subunit Mis12 complex contains Mis12, Pmf1, Nsl1 and Dsn1 [192,193]. The budding yeast equivalent of this complex is the MIND (or Mtw1) complex [194,195]. The Mis12 complex interacts with both CENP-C and CENP-T, serving as an important interaction hub between the KMN assembly and the inner kinetochore [196,197]. The complex also provides a binding site for Knl1 and the Ndc80 complex [173,192,198]. As discussed above, CENP-T directly binds up to two Ndc80 complexes. CENP-T also binds Mis12, which then interacts with

Ndc80, providing a mechanism by which a third Ndc80 complex can be indirectly recruited by a single CENP-T complex [174]. Since Mis12 also interacts with CENP-C, it has been proposed that two Mis12 complexes could be recruited to by a single CCAN, making it possible for one CCAN to ultimately bind up to four Ndc80 complexes [161,174]. The Mis12 complex is roughly rod-shaped, with high helical content as well as linear motifs (figure 6e) [193,194,198].

## 13. Knl1 complex

The Knl1 complex is composed of Knl1 and ZWINT. Knl1 is the largest structural subunit in the outer kinetochore. Its structure is largely intrinsically disordered with the exception of the C-terminal approximately 500 a.a. residues [198]. The C-terminal ordered region contains a coiled coil followed by paired RWD domains, a structure reminiscent of the other kinetochore proteins Scp24, Spc25, CENP-O and CENP-P [162,186,198]. The RWD domains of Knl1 mediate a direct interaction with the stalk of the Mis12 complex (figure 6f), and Knl1 also binds many other proteins that associate with the outer kinetochore [198]. ZWINT, the other component of the Knl1 complex, plays a central role in the spindle assembly checkpoint (SAC), an essential pathway that ensures proper chromosome alignments and kinetochore–microtubule attachments before proceeding with mitosis [199–203].

## 14. Connections to microtubules

As discussed above, the Ndc80 complex represents a central component in mediating kinetochore–microtubule interactions. However, several other proteins can make these connections and/or physically support these interactions. Here we will highlight a few key players.

The Ska complex, a trimer of Ska1, Ska2 and Ska3, functions to enhance the microtubule binding ability of the Ndc80 complex [204–206]. The Ska complex has a 'W'-like shape and is composed of dimers of helical bundles of its subunits (figure 6g) [206]. The C-terminal winged-helix domain of Ska1 interacts with tubulin and, unlike the Ndc80 complex, indiscriminately tracks the tips of both polymerizing and depolymerizing microtubules [207–209]. The unstructured C-terminal region of Ska3 also facilitates interaction between the complex and microtubules [207]. The budding yeast equivalent of the Ska complex appears to be the Dam1 complex [210–213], although the two complexes have no structural similarity. Beyond the Ska complex, multiple proteins including dynein, kinesin, CENP-E and CENP-F have been shown to form weak attachments to microtubules in the absence of the Ndc80 complex [214–216], suggesting that they may also be involved in kinetochore–microtubule interactions.

A striking feature of the metazoan regional kinetochore is that it expands into a crescent-like shape prior to end-on microtubule binding by the Ndc80 complex (figure 1b) [10,11,217]. This process involves numerous molecules including CENP-E, CENP-F, the dynein/dynactin motor complex and the ROD–ZW10–Zwilch (RZZ) complex [11,214,218,219]. This physical transformation of the kinetochore may serve to increase the likelihood of microtubule capture and may also promote SAC signalling [11]. Once end-on microtubule attachments to the Ndc80 complex are established, the kinetochore retracts from this expanded crescent shape into a smaller plate [11,220]. This retraction of the kinetochore involves the release of the dynein/dynactin and RZZ complexes along with the central SAC complex, halting SAC signalling at that kinetochore [221–227].

Many additional proteins play important roles in forming, maintaining and regulating connections at the kinetochore. These include the SKAP-Astrin complex (which regulates proper kinetochore–microtubule attachments [228–232]), XMAP215 (which is involved in microtubule polymerization and nucleation [233–237]), the CLASPs (which regulate microtubule dynamics [238–241]), MCAK (which is a microtubule depolymerase [242–246]), and more. Discussion of these proteins is beyond the scope of this review, but the kinetochore–microtubule interface has been reviewed elsewhere in depth [247].

## 15. Putting the pieces together—models of the chromatin-to-microtubule connection

Because of the overall symmetry of the octameric CENP-A nucleosome, harbouring two histone surfaces or 'faces' for equivalent recruitment of the recognition subunits of the CCAN (i.e. CENP-C and CENP-N that form the CCNC), the most prominent models for the chromatin connection to microtubules have generally featured a stoichiometry of two CCAN complexes per CENP-A nucleosome (figure 7, model 1) [5]. This has been supported by the sedimentation behaviour of reconstituted particles containing a large subset of recombinant CCAN components [139]. An alternative model has emerged, though, where a single CCAN connects the CENP-A nucleosome to the outer kinetochore (figure 7, model 2) [86,144]. The evidence for the alternate model comes from structural work that strongly implies that the sequence of the underlying DNA can influence stoichiometry and the fidelity of assembly of the CCNC [86,135,136,144] and observations of cell cycle changes in the stoichiometry within the CCNC (i.e. losing one copy of CENP-N at mitotic onset) [86]. Since both CENP-C and CENP-N within the CCNC are required for the assembly of the CCAN, it is likely that both constituents are present on the face of the nucleosome that assembles the CCAN. Analysis of the composition of tagged centromere proteins in mammalian cells have indeed reported that CENP-C is super-stoichiometric to at least one CCAN subunit, CENP-T [248]. Further investigation of the stoichiometry of individual sub-complexes of the CCAN at the onset of mitosis will reveal whether or not there are other mechanisms to adjust its composition at the time of kinetochore assembly.

The two models are primarily presented here to interrogate the implications of the orientation of CENP-A nucleosomes at the interface between the chromosome and the cytosolic surface where microtubules attach. Microtubule attachment is at a perpendicular orientation to the chromosome (typically termed 'end-on'). The models are not intended to predict differences in the stoichiometry of outer kinetochore components, since there are many ways in which their stoichiometry could be modulated (e.g. expression levels limiting a single connector component or homo-oligomerization of a sub-complex that amplifies some subunits). Further, either model can readily accommodate prior subunit distance measurements made at natural mitotic kinetochores [152,249].

Regarding CENP-A nucleosome orientation, in model 1, the two histone surfaces are equivalently used to recruit an

royalsocietypublishing.org/journal/rsob    Open Biol. **10**: 200051

royalsocietypublishing.org/journal/rsob    Open Biol. **10**: 200051

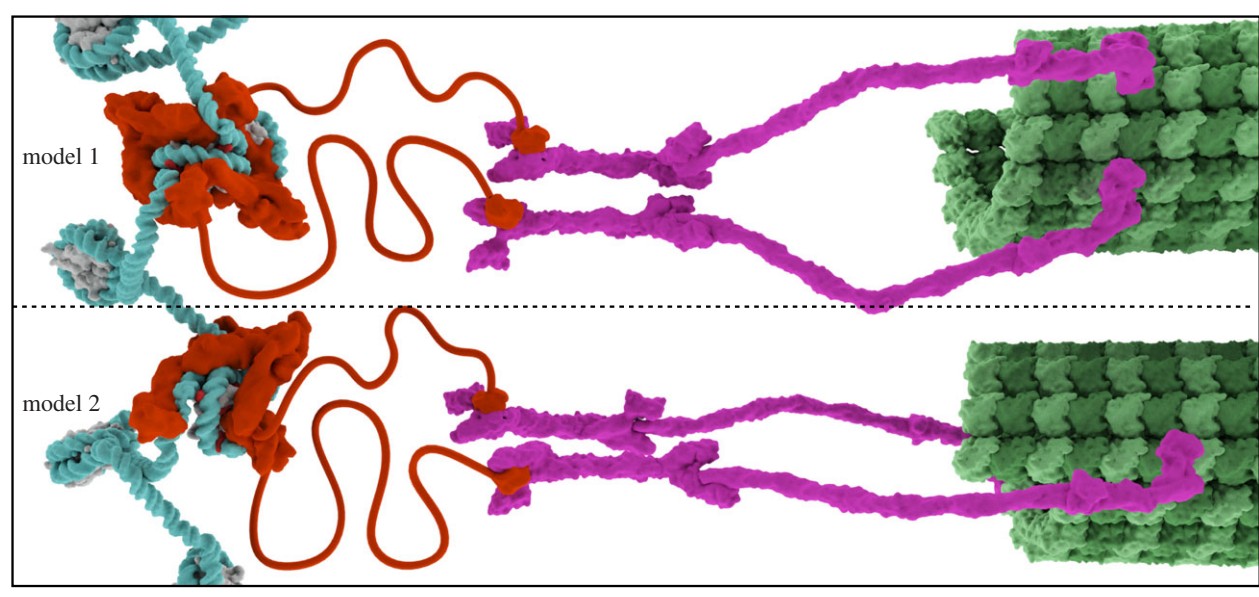

**Figure 7.** Models of the kinetochore organization on centromeric chromatin. Model 1 has a symmetric arrangement of two copies of the CCAN bound per CENP-A nucleosome, whereas model 2 has an asymmetric arrangement with a single copy of the CCAN per CENP-A nucleosome. Both models contain two copies of CENP-C per CENP-A nucleosome. The proposed connection through CENP-C to neighbouring nucleosomes has been omitted from both models for visual clarity. See text for details about the two models, the data supporting each one and implications for centromeric chromatin structure during mitotic chromosome segregation.

entire CCAN, and the orientation is a symmetric one. One could imagine several orientations of the CENP-A nucleosome, and we illustrate it with the dyad rotated 90° from the microtubule. This would have the entry/exit DNA from the CENP-A nucleosome slightly peeling off the histone octamer and proceeding towards the neighbouring nucleosomes. A dyad position towards the microtubule is more difficult to imagine because this would be the same surface as the exit/entry DNA. Connection to neighbouring nucleosomes, in this case, could be accomplished by a greater amount of DNA unwrapping from the CENP-A nucleosome core, however. In model 2, the histone surfaces are not equivalent. On the surface facing the microtubule, a full CCAN complex is assembled and this nucleates the kinetochore. On the opposite surface, facing the chromosome (i.e. mainly abutting conventional nucleosomes containing histone H3), the full CCAN is absent, but a second copy of CENP-C is present. One potential outcome of this orientation is to use the copy of CENP-C on the chromosome face (and not engaged with the CCAN) to physically connect CENP-A nucleosomes to neighbouring nucleosomes. These connections could help rigidify CENP-A chromatin under spindle pulling forces, since CENP-A chromatin does not substantially deform under these forces, whereas the intervening chromatin at the inner centromere is substantially elongated at mitosis [250].

Another question that must be resolved is how well conserved subunit stoichiometry and arrangement at the centromere are between species. On one hand, one might argue that since the process of chromosome segregation is so fundamental to eukaryotic life that there must be conservation of structure even among diverse species (e.g. budding yeast and humans). From this viewpoint, the surprising arrangement of CENP-L$^{Iml3}$-CENP-N$^{Chl4}$ away from the histone surface of CENP-A in the recent CENP-A nucleosome CCAN/Ctf19 structure (figure 4c) [144] (in the face of strong evidence from metazoan systems that CENP-N binds elsewhere; figure 3d–f) might lead one to question the validity of the structure, itself. On the other hand, we know that there is rampant evolutionary innovation at the centromere [251]. One example of this is the DNA

sequence defined elements in budding yeast that recruit CBF1 and CBF3 complexes [252,253]. Another is the recurrent complete loss of CENP-A nucleosomes (and many other CCAN components) in some insect lineages [254]. From that point of view, an innovation like repositioning CENP-L$^{Iml3}$-CENP-N$^{Chl4}$ is not so difficult to imagine.

Regarding the impact of DNA sequence, the experimental findings come primarily from comparing CENP-A nucleosomes assembled with natural centromere DNA to those assembled with a synthetic nucleosome positioning sequence used nearly ubiquitously in chromatin structural biochemistry, the Widom '601' sequence [149], as mentioned above. Cryo-EM analysis of CENP-A nucleosomes assembled with the 601 sequence and bound by CENP-N initially revealed high-resolution structural models with a form of the complex containing a single copy of CENP-N [135,136]. Assembly of CENP-A nucleosomes on α-satellite DNA leads to robust formation of the CCNC with two copies each bound of CENP-C and CENP-N, relative to those assembled with the 601 DNA sequence [86]. This led to single-particle populations yielding high-resolution cryo-EM structures of forms containing two copies of CENP-C and either one or two copies of CENP-N [86]. The AT-rich nature of natural centromeric DNA imparts flexibility that the strong nucleosome positioning 601 sequence lacks, permitting bends in the DNA and a DNA path that creates symmetrical binding sites on each side of the nucleosome. In yeast, centromeric DNA is extremely AT-rich (greater than 85%), and, as mentioned above, this has been reported to affect the stoichiometry of the other CCNC component, CENP-C [144]. Only a single copy of CENP-C$^{Mif2}$ binds CENP-A nucleosomes assembled on 601 DNA [144], whereas two copies of CENP-C$^{Mif2}$ bind CENP-A assembled on yeast CEN DNA [133,144]. It is tempting to speculate that some natural DNA sequences could be refractory to centromere formation because, like the artificial 601 sequence, they do not accommodate the assembly of the CCAN starting with CENP-C and/or CENP-N. These effects certainly not all-or-none, however, since solution biochemistry experiments

supporting model 1 (figure 7) described above were performed using the 601 sequence [139], and the cryo-EM experiment on 601 DNA described above also identified a minority of particles containing two copies of CENP-N, a fraction which increased in the presence of excess CENP-N [136]. Analysis of the high-resolution 601 DNA CENP-A nucleosome structures reveals marked asymmetry of the two histone surfaces of the nucleosome [86]. It is most likely that the assembly and stoichiometry of the CCAN is affected by a variety of factors, including DNA sequence, but also reconstitution conditions and variation among species. The field must continue to investigate CCAN assembly and structure *in vitro*, but must also meet the challenge of devising compelling ways to test the hypotheses emerging from these studies on natural centromeres in cells.

Regarding the cell cycle regulation, the relative abundance of CENP-N to CENP-C drops at centromeres immediately at the onset of mitosis [86]. Since both CENP-C and CENP-N are required for the recruitment of other CCAN components to CENP-A nucleosomes [26,81–83,85,130,135,136,150], and since CENP-A nucleosomes do not change in number upon mitotic entry [101], this finding suggests that the number of CCAN complexes per unit of centromeric chromatin drops by half prior to formation of the mitotic kinetochore. The model with a single CCAN per CENP-A nucleosome (figure 7, model 2) is built upon the assumption that most individual constitutive centromere components assemble into discrete CCAN complexes bound to the CENP-A nucleosome. The extensive contacts across the majority of CCAN components evident from the recent structural advances (figures 4 and 5) [86,144–146,148] support the notion that the CCAN can exist as a discrete unit. This does not rule out the possibility that CCAN stoichiometry can be modulated in important ways, and indeed there is evidence that binding surfaces with the CCAN can be rearranged in mitosis [131,255].

Future efforts will be required to differentiate between existing models (figure 7) and will also probably lead to discovery of other important ways in which the CCAN is assembled and modulated as it is in the process of nucleating kinetochore formation.

# 16. Outlook

Proper assembly of the kinetochore at the centromere is a fundamental process of cell biology that is crucial for accurate propagation of genetic information through cell and organismal generations. However, in spite of recent landmark achievements in the structural understanding of the centromere and kinetochore, there remains much to be learned about the structure, organization and function of the centromere and kinetochore components that make this process possible. Determining the precise mechanism of the deposition of CENP-A nucleosomes represents an important challenge for the field. Furthermore, the most comprehensive structures for inner kinetochore components come from budding yeast, and although many yeast kinetochore proteins share homology with their human counterparts, structures of centromere complexes from other species, including from metazoan species, will be critical. They will help to develop an understanding of which structural aspects are universal in eukaryotes and which are undergoing innovations that accompany chromosome evolution. As with essentially all macromolecular complexes, connecting structures of purified/reconstituted complexes to the functionally important structures that exist *in vivo* represents a major hurdle, but one that is crucial to pursue, especially since the CCAN undergoes changes depending on cell cycle state. Technological developments continue to allow us to study structures at higher resolution and in more native contexts, promising a period in the coming years of exciting progress to reveal precisely how the kinetochore connects centromeric chromatin to spindle microtubules (table 1).

Data accessibility. This article has no additional data.

Authors' contributions. All of the authors wrote the review.

Competing interests. We declare we have no competing interests.

Funding. This work was supported by NIH grant no. GM130302 (B.E.B.).

Acknowledgements. We thank the anonymous peer reviewers for their suggestions to improve our manuscript.

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
