## [Reviewer comments · Open Biology]

Review History

RSOB-20-0051.R0 (Original submission)

Review form: Reviewer 1

Recommendation

Accept with minor revision (please list in comments)

Do you have any ethical concerns with this paper?

No

Comments to the Author

This manuscript by Kixmoeller et al., does a great job of integrating all available molecular and structural level information on molecular players involved in centromere maintenance and kinetochore assembly, fundamental processes essential for cell and organismal viability. I am confident that this review article will be a valuable resource to anyone interested in understanding the fundamental inner workings of processes controlling cell survival in general and cell division in particular. This article will also be appreciated by structural biologists interested in understanding the structural basis of macromolecular assemblies and function.

Overall the manuscript is well written. However, I have a few suggestions that the authors might

want to consider to make it even more effective.

- The following works are not mentioned/discussed and need to be included to make this manuscript complete.
 - Recent structural work on Drosophila Cal1 interaction with CENP-A(Cid)/H4 and CENP-C cupin domain interaction (Medina-Pritchard et al., EMBO J 2020) needs to be discussed both in the context of CENP-A/H4 recognition and centromere targeting.
 - Where Ndc80 phosphorylation is discussed, Ndc80 phosphorylation by Aurora A kinase also needs to be mentioned (DeLuca et al., JCB 2018)
 - Where the Ska complex is discussed, structural works on the Ska1 microtubule binding domains are not mentioned or cited – (Abad et al., Nature Communications 2014; Monda et al., Current Biology 2017). The work by Abad et al., provided the structural basis for how the Ska complex (unlike the Ndc80 complex) interacts with polymerising and depolymerising microtubules in an indiscriminative manner.

- Manuscript will benefit by careful re-reading to look for grammatic errors, format consistency and convoluted sentences.

Examples:

Page 2, Introduction, first paragraph, line 3 - ...replicated and segregated during every cell cycle ('during' missing);

Page 3, Paragraph 3, line 2 - where Histone H3 is introduce, better to explicitly introduce as Histone H3 rather than H3;

Page 3, Paragraph 3, line 7 - CCAN - Constitutive Centromere ... (not Constitute);

Page 3, last paragraph, line 2 - ..and in organisms (I guess, it has to be 'model organisms'); Page 4, last sentence might benefit from rephrasing;

Page 7, second paragraph, HJURP is sufficient *to* incorporation.....; (sufficient for?)

Page 11, paragraph 3, line 5 - superscript missing for CENP-I ;

Page 11, paragraph 3, line 5 from bottom - '...binding cite...' - it has to be 'binding site'

Page 15, paragraph 1, line 7 - rephrasing the sentence starting 'Supporting the importance...' might help;

Page 15, penultimate sentence - '...CCAN, making it possible for one CCAN to ultimately bind up to four...' (I guess 'bind' missing)

Several places, where proximity of the domain to the C or N terminus is described 'terminal' is better suited than 'terminus' - for ex: The C-terminal region or the C-terminal domain.

Page 17, paragraph 3, line 6 - '...reconstituted particles containing *with* a large subset ...'

Page 17, line 4 from bottom, '...DNA leads to more faithful assembly...' - I am not convinced that the use of 'faithful' here is appropriate.

Page 18, line 5, rephrasing the sentence starting 'On CENP-A nucleosomes...' might benefit.

- Where CENP-A CATD introduced, it would help to include a panel showing the sequence alignment with secondary structure elements and the domain
- It would help to use arrows in the figures wherever possible, to highlight features that the readers are expected to focus. This will particularly help those general readers who are not very familiar looking at protein structures. Labeling domains (such as the CH domain, RWD domain) in the figures will also help. It would also help to include the domain architecture of CENP-C in the figure, where it is discussed.
- Inner and outer kinetochore subcomplexes are shown in similar colors - it would help if contrasting colours can be chosen.
- In the figure legends, where possible organism name needs to be included - if the structure comes from pombe, cerevisiae or human.

Review form: Reviewer 2

Recommendation

Major revision is needed (please make suggestions in comments)

Do you have any ethical concerns with this paper?

No

Comments to the Author

This review by Kixmoeller and coworkers focuses on the kinetochore, a large macromolecular complex that bridges chromosome and microtubules during mitosis to the segregation of chromosomes in the daughter cells. There has been considerable recent progress in this area, fueled by efforts of biochemical reconstitution and structural analysis. A review on the topic is therefore timely and has the potential to occupy an empty niche.

The review aims to be comprehensive. This is not easy, because kinetochores are very large structures. In its present form, the review focuses more closely on the so-called CCAN, the inner kinetochore complex. The outer kinetochore and its control of microtubule binding is treated rather superficially and one wonders if it should be covered at all at this superficial level. It leaves a feeling of incompleteness.

With concern to CCAN, on the other hand, the review covers individual interactions between kinetochore protein complexes in some detail, therefore making it a good summary for people approaching the field. On the other hand, the figures lack detail on the individual protein complexes, and are difficult to read. The manuscript would benefit considerably if figures are improved.

Another issue that the authors should consider is that the review falls short of making critical sense of the implications of two models being proposed based on current structural information. This is detailed below. In my opinion, models should try to explain something, i.e. should be clearly distinct from the observations that support them, and it should be clear what their value is. This is not what happened here.

Finally, there are problems with the cited literature, some of which are listed below.

In summary, I praise the authors for their effort. At the same time, I feel rather lukewarm about this review and what it tries to achieve. I hope that the following comment will direct the authors towards assembling an improved review.

Specific points

The choice of cited literature is absolutely crucial in a review and conveys a sense of how authoritative and influential the review is going to be. Here it is "O.K-ish". It could be improved to remove inaccuracies and inconsistencies. Here are a few examples, but the problem is more pervasive.

"Structures are available for each individual component bound or in combination (Figure 3b-e) [67,68,72,108]." The choice of literature supporting this statement is incomplete. The authors should specifically mention the work of Chittori et al. paper here and also cite Tian et al.:

Tian T, Li X, Liu Y, Wang C, Liu X, Bi G, Zhang X, Yao X, Zhou ZH, Zang J. Molecular basis for CENP-N recognition of CENP-A nucleosome on the human kinetochore. *Cell Res.* 2018 Mar;28(3):374-378. doi: 10.1038/cr.2018.13.

“Other crystallographic studies have reported the structures of CENP-M (Figure 4c) [111]” Reference 111, Weir et al. 2016, does not report the structure of CENP-M. That was rather described in Basilico et al. 2014 (reference 126).

“Both CENP-C and CENP-N are required to recruit all other CCAN components [23,63,64,65,108,109,118,121,122].” Why is reference 121 part of this group?

“The complex is composed of two heterodimers, Ndc80-Nuf2, and Spc24-Spc25 [143,149,150].” These are important references but they do not make that point, which is rather the focus of references 151-153 cited two lines later. What about John Kilmartin’s work? He was the one showing Ndc80 is a tetramer.

“Supporting the importance of cooperative binding is evidence that, unlike individual Ndc80 complexes in solution, Ndc80 complexes immobilized at high concentration on beads can create load-bearing attachments to depolymerizing microtubules [2,26,159,160].” Here the authors cite three reviews and a single research paper. The reviews do not focus at all on the issue of cooperative binding. The research paper has one experiment on this, but the issue has been the subject of more recent work (e.g from the labs of De Luca, Grishchuk, and Musacchio). If the authors do not want to cite original literature, they could elect to cite the 2020 Wimbish and DeLuca excellent review that is entirely dedicated to this topic.

“The 4-subunit Mis12 complex contains Mis12, Pmf1, Nsl1, and Dsn1 [162]. The budding yeast equivalent of this complex is the MIND (or Mtw1) complex [163].” Again, this is an incorrect choice of literature in this context. Both references describe analyses of the human complex, not of the yeast complex. The structure of the yeast complex was described by Dimitrova et al., reference 166.

“The Mis12 complex interacts with both CENP-C and CENP-T, serving as an important interaction hub between the KMN assembly and the inner kinetochore [164].” Reference 164 describes the interaction of MIS12 with CENP-C, but the interaction with CENP-T was described by Gascoigne et al. in humans and Malvezzi et al. in yeast.

“This retraction of the kinetochore involves the release of the dynein/dynactin and RZZ complexes along with the central SAC complex, halting SAC signaling at that kinetochore [180].” Reference 180 is excellent but it focuses on Spindly, a protein that is not even included in the authors’ list of corona component. The role of RZZ in checkpoint signaling was established in the first decade of this millennium in the Salmon and Karess labs.

There are several of these shortcomings and I would therefore urge the authors to go through the review and try to remove these inaccuracies and inconsistencies.

Page 6

The second half of this page is devoted to the description of the loose ends of the CENP-A nucleosome, super-helical bulges, and to whatever else is different relative to H3 nucleosomes, stressing that the structural changes are important for CCAN assembly. All this comes across as a reflection of the authors’ emphasis on their own work, and they can hardly be blamed for it as this is their review. My personal opinion, however, is that in the absence of structures of the human CCAN-CENP-A nucleosome complex, the importance of these effects cannot be ascertained with any degree of approximation. I would exercise more caution and state that in the absence of further evidence, an assessment of the importance of these structural features, which have emerged from greatly incomplete, low-resolution structures, must be taken with a grain of salt. More on this below.

The description of the connections to microtubules on page 16 is very sketchy and largely limited in scope. I am not sure how it could be improved, but the authors could at least indicate a full list of proteins involved, including at least the SKAP-Astrin complex, XMAP215, the CLASPs, and

MCAK, which at present are not even cited. A table or an extension of a figure could be used for this. In general, I don't see that citing a tiny amount of the original literature and resorting to other people's reviews is a very useful exercise. One may well remove this part and clearly state that it is not the focus of the review.

I have many problems with the discussion of the two models at the end of the review. In my opinion, this final section lacks critical thinking, it is confusing, and will be hard to follow for most readers.

First of all, there should be appropriate coverage of all the evidence on the stoichiometry of kinetochore assembly. For instance, the authors should cite the work of Suzuki et al. in Nature Communications on the stoichiometry of kinetochore subunits.

Second, the authors write: "On CENP-A nucleosomes assembled on 601 DNA only a single copy of CENP-C binds [117], while when assembled on yeast CEN DNA two copies of CENP-C binds [105,117]. It is tempting to speculate that some natural DNA sequences could be refractory to centromere formation because like the artificial 601 sequence they do not accommodate the assembly of the CCAN starting with CENP-C and/or CENP-N". Earlier in the section, they also write that "the most prominent models for the chromatin connection to microtubules have generally featured a stoichiometry of two CCAN complexes per CENP-A nucleosome (Figure 6, Model 1) [136]. This has been supported by the sedimentation behavior of reconstituted particles containing with a large subset of recombinant CCAN components [111]." These sedimentation experiments were carried out with nucleosomes build on the 601 sequence. Does that fit the authors' narration?

Third, it is fine to discuss how the choice of DNA may influence the biochemistry of the system, but the discussion should focus on more impending questions. First, the authors should clarify whether the structure of the complex by the Barford laboratory is a reasonable account of what we know of the yeast kinetochore in the first place. How sensible is it to assume that CENP-N occupies a different position in the yeast structure relative to that of humans? How strong is the conservation of interfaces that would support functional similarities or differences? What could have gone wrong in structure determination, if anything at all (e.g. what fraction of particles out of the total were used in the reconstructions)? What are the implications of removing half of CENP-N before mitotic entry? How can the structure accommodate such a large structural change given what has been discussed before on the foundational role of CENP-N for the stability of the inner kinetochore? How could these models be tested? Importantly: an observation is not a model. Models have to try to explain something, else what are they good for? As far as model 2 is concerned, there are some clear predictions, including that the stoichiometry of other inner and outer kinetochore subunits will most likely change if half of CENP-N falls off. Has this been observed, does it fit what we know (check the Suzuki et al. paper, for instance)? Just to clarify how problematic model 2 is, the authors should consider that in *S. cerevisiae* Cnn1/CENP-T is not essential. Model 2 predicts that in cells lacking Cnn1 there will be a single functional Ndc80 tetramer (the one bound to the single Mif2/CENP-C). How realistic is that? What are the implications regarding what we know about cooperativity of NDC80 binding to microtubules? These are the type of implications that a review should discuss if it aims to have a durable impact on a field.

Finally, a note on figures. Fig. 4 shows almost everything in orange. The authors should use at least some shades of orange-reddish colors to distinguish the different chains, otherwise these diagrams won't be useful. That the orientation of Fig. 4f is different from Fig. 4 g-j is also sub-optimal. The final model in Fig. 6 could be greatly improved. What is the evidence of a "solid" connection from Ndc80 to the nucleosome, without any unstructured connecting parts where CENP-C and CENP-T? A schematic of CENP-C and CENP-T with their interacting motifs would be helpful. And a fair representation of the stoichiometries is essential.

Other points in no specific order

Incomplete sentence: “one CCAN to ultimately up to four Ndc80”

“The C-terminus ordered region contains a coiled coil followed by paired RWD domains...” “C-terminus” is a name, the adjective is C-terminal.

“Knl1 also contains multiple binding sites for proteins that associate with the outer kinetochore, including ZWINT” ZWINT is a core component of the 2-subunit KNL1 complex. It should be included in the scheme in Figure 1.

“reconstituted particles containing with a large subset of recombinant CCAN components”

Why are references coming with all authors or only the first author?

Tanaka K, Chang HL, Kagami A, Watanabe Y. 2009 CENP-C functions as a scaffold for effectors with essential kinetochore functions in mitosis and meiosis. *Dev. Cell* 17, 334– 343. (doi:10.1016/j.devcel.2009.08.004)

Xiao H et al. 2017 Molecular basis of CENP-C association with the CENP-A nucleosome at yeast centromeres. *Genes Dev.* 31, 1958–1972. (doi:10.1101/gad.304782.117)

The description of centromere sequences is rather sketchy. A diagram of the human centromere sequence would also help (i.e. a more detailed schematic to replace Fig. 2A).

“New incorporation of CENP-A happens in G1, after mitotic exit, whereas new H3 is incorporated during DNA replication [82,85]”. Here the authors should cite the paper by Elaine Dunleavy and Gary Karpen as well (Dunleavy et al., 2011 – Nucleus).

For the sections CENP-C and CENP-LN , CENP-HIKM, CENP-TWSX , CENP-OPQUR, it would be helpful to add more detailed figures of the complexes described here. There is a lot of descriptive information that will not be digested easily by non-structural biologists (and a review such this, should be a useful compendium for them). For example, Figure 4 could be expanded to have single cartoons of the complexes, clearly labelled to visually elucidate the corresponding text.

Decision letter (RSOB-20-0051.R0)

14-Apr-2020

Dear Dr Black,

We are writing to inform you that the Editor has reached a decision on your manuscript RSOB-20-0051 entitled "The Kinetochore Comes into Focus – From Centromeric Chromatin to Microtubule Connections", submitted to Open Biology.

As you will see from the reviewers’ comments below, there are a number of criticisms that prevent us from accepting your manuscript at this stage. The reviewers suggest, however, that a revised version could be acceptable, if you are able to address their concerns. If you think that you can deal satisfactorily with the reviewer’s suggestions, we would be pleased to consider a revised manuscript.

The revision will be re-reviewed, where possible, by the original referees. As such, please submit

the revised version of your manuscript within four weeks. If you do not think you will be able to meet this date please let us know immediately.

When submitting your revised manuscript, please respond to the comments made by the referee(s) and upload a file "Response to Referees" in "Section 6 - File Upload". You can use this to document any changes you make to the original manuscript. In order to expedite the processing of the revised manuscript, please be as specific as possible in your response to the referee(s).

Please see our detailed instructions for revision requirements
<https://royalsociety.org/journals/authors/author-guidelines/>

Sincerely,
 The Open Biology Team
 mailto: openbiology@royalsociety.org

Reviewer(s)' Comments to Author(s):
 Referee: 1

Comments to the Author(s)

This manuscript by Kixmoeller et al., does a great job of integrating all available molecular and structural level information on molecular players involved in centromere maintenance and kinetochore assembly, fundamental processes essential for cell and organismal viability. I am confident that this review article will be a valuable resource to anyone interested in understanding the fundamental inner workings of processes controlling cell survival in general and cell division in particular. This article will also be appreciated by structural biologists interested in understanding the structural basis of macromolecular assemblies and function.

Overall the manuscript is well written. However, I have a few suggestions that the authors might want to consider to make it even more effective.

- The following works are not mentioned/discussed and need to be included to make this manuscript complete.
 - Recent structural work on Drosophila Cal1 interaction with CENP-A(Cid)/H4 and CENP-C cupin domain interaction (Medina-Pritchard et al., EMBO J 2020) needs to be discussed both in the context of CENP-A/H4 recognition and centromere targeting.
 - Where Ndc80 phosphorylation is discussed, Ndc80 phosphorylation by Aurora A kinase also needs to be mentioned (DeLuca et al., JCB 2018)
 - Where the Ska complex is discussed, structural works on the Ska1 microtubule binding domains are not mentioned or cited - (Abad et al., Nature Communications 2014; Monda et al., Current Biology 2017). The work by Abad et al., provided the structural basis for how the Ska complex (unlike the Ndc80 complex) interacts with polymerising and depolymerising microtubules in an indiscriminative manner.

- Manuscript will benefit by careful re-reading to look for grammatic errors, format consistency and convoluted sentences.

Examples:

Page 2, Introduction, first paragraph, line 3 - ...replicated and segregated during every cell cycle ('during' missing);

Page 3, Paragraph 3, line 2 - where Histone H3 is introduce, better to explicitly introduce as Histone H3 rather than H3;

Page 3, Paragraph 3, line 7 - CCAN - Constitutive Centromere ... (not Constitute);

Page 3, last paragraph, line 2 - ..and in organisms (I guess, it has to be 'model organisms'); Page 4, last sentence might benefit from rephrasing;

Page 7, second paragraph, HJURP is sufficient *to* incorporation.....; (sufficient for?)

Page 11, paragraph 3, line 5 - superscript missing for CENP-I ;

Page 11, paragraph 3, line 5 from bottom - '...binding cite...' - it has to be 'binding site'

Page 15, paragraph 1, line 7 - rephrasing the sentence starting 'Supporting the importance...' might help;

Page 15, penultimate sentence - '...CCAN, making it possible for one CCAN to ultimately bind up to four...' (I guess 'bind' missing)

Several places, where proximity of the domain to the C or N terminus is described 'terminal' is better suited than 'terminus' - for ex: The C-terminal region or the C-terminal domain.

Page 17, paragraph 3, line 6 - '...reconstituted particles containing *with* a large subset ...'

Page 17, line 4 from bottom, '...DNA leads to more faithful assembly...' - I am not convinced that the use of 'faithful' here is appropriate.

Page 18, line 5, rephrasing the sentence starting 'On CENP-A nucleosomes...' might benefit.

- Where CENP-A CATD introduced, it would help to include a panel showing the sequence alignment with secondary structure elements and the domain
- It would help to use arrows in the figures wherever possible, to highlight features that the readers are expected to focus. This will particularly help those general readers who are not very familiar looking at protein structures. Labeling domains (such as the CH domain, RWD domain) in the figures will also help. It would also help to include the domain architecture of CENP-C in the figure, where it is discussed.
- Inner and outer kinetochore subcomplexes are shown in similar colors - it would help if contrasting colours can be chosen.
- In the figure legends, where possible organism name needs to be included - if the structure comes from pombe, cerevisiae or human.

Referee: 2

Comments to the Author(s)

This review by Kixmoeller and coworkers focuses on the kinetochore, a large macromolecular complex that bridges chromosome and microtubules during mitosis to the segregation of chromosomes in the daughter cells. There has been considerable recent progress in this area, fueled by efforts of biochemical reconstitution and structural analysis. A review on the topic is therefore timely and has the potential to occupy an empty niche.

The review aims to be comprehensive. This is not easy, because kinetochores are very large structures. In its present form, the review focuses more closely on the so-called CCAN, the inner kinetochore complex. The outer kinetochore and its control of microtubule binding is treated rather superficially and one wonders if it should be covered at all at this superficial level. It leaves a feeling of incompleteness.

With concern to CCAN, on the other hand, the review covers individual interactions between kinetochore protein complexes in some detail, therefore making it a good summary for people approaching the field. On the other hand, the figures lack detail on the individual protein complexes, and are difficult to read. The manuscript would benefit considerably if figures are improved.

Another issue that the authors should consider is that the review falls short of making critical sense of the implications of two models being proposed based on current structural information. This is detailed below. In my opinion, models should try to explain something, i.e. should be clearly distinct from the observations that support them, and it should be clear what their value is. This is not what happened here.

Finally, there are problems with the cited literature, some of which are listed below.

In summary, I praise the authors for their effort. At the same time, I feel rather lukewarm about this review and what it tries to achieve. I hope that the following comment will direct the authors towards assembling an improved review.

Specific points

The choice of cited literature is absolutely crucial in a review and conveys a sense of how authoritative and influential the review is going to be. Here it is "O.K-ish". It could be improved to remove inaccuracies and inconsistencies. Here are a few examples, but the problem is more pervasive.

"Structures are available for each individual component bound or in combination (Figure 3b-e) [67,68,72,108]." The choice of literature supporting this statement is incomplete. The authors should specifically mention the work of Chittori et al. paper here and also cite Tian et al.:

Tian T, Li X, Liu Y, Wang C, Liu X, Bi G, Zhang X, Yao X, Zhou ZH, Zang J. Molecular basis for CENP-N recognition of CENP-A nucleosome on the human kinetochore. *Cell Res.* 2018 Mar;28(3):374-378. doi: 10.1038/cr.2018.13.

"Other crystallographic studies have reported the structures of CENP-M (Figure 4c) [111]" Reference 111, Weir et al. 2016, does not report the structure of CENP-M. That was rather described in Basilico et al. 2014 (reference 126).

"Both CENP-C and CENP-N are required to recruit all other CCAN components [23,63,64,65,108,109,118,121,122]." Why is reference 121 part of this group?

"The complex is composed of two heterodimers, Ndc80-Nuf2, and Spc24-Spc25 [143,149,150]." These are important references but they do not make that point, which is rather the focus of references 151-153 cited two lines later. What about John Kilmartin's work? He was the one showing Ndc80 is a tetramer.

"Supporting the importance of cooperative binding is evidence that, unlike individual Ndc80 complexes in solution, Ndc80 complexes immobilized at high concentration on beads can create load-bearing attachments to depolymerizing microtubules [2,26,159,160]." Here the authors cite three reviews and a single research paper. The reviews do not focus at all on the issue of cooperative binding. The research paper has one experiment on this, but the issue has been the subject of more recent work (e.g from the labs of De Luca, Grishchuk, and Musacchio). If the authors do not want to cite original literature, they could elect to cite the 2020 Wimbish and DeLuca excellent review that is entirely dedicated to this topic.

"The 4-subunit Mis12 complex contains Mis12, Pmf1, Nsl1, and Dsn1 [162]. The budding yeast

equivalent of this complex is the MIND (or Mtw1) complex [163].” Again, this is an incorrect choice of literature in this context. Both references describe analyses of the human complex, not of the yeast complex. The structure of the yeast complex was described by Dimitrova et al., reference 166.

“The Mis12 complex interacts with both CENP-C and CENP-T, serving as an important interaction hub between the KMN assembly and the inner kinetochore [164].” Reference 164 describes the interaction of MIS12 with CENP-C, but the interaction with CENP-T was described by Gascoigne et al. in humans and Malvezzi et al. in yeast.

“This retraction of the kinetochore involves the release of the dynein/dynactin and RZZ complexes along with the central SAC complex, halting SAC signaling at that kinetochore [180].” Reference 180 is excellent but it focuses on Spindly, a protein that is not even included in the authors’ list of corona component. The role of RZZ in checkpoint signaling was established in the first decade of this millennium in the Salmon and Karess labs.

There are several of these shortcomings and I would therefore urge the authors to go through the review and try to remove these inaccuracies and inconsistencies.

Page 6

The second half of this page is devoted to the description of the loose ends of the CENP-A nucleosome, super-helical bulges, and to whatever else is different relative to H3 nucleosomes, stressing that the structural changes are important for CCAN assembly. All this comes across as a reflection of the authors’ emphasis on their own work, and they can hardly be blamed for it as this is their review. My personal opinion, however, is that in the absence of structures of the human CCAN-CENP-A nucleosome complex, the importance of these effects cannot be ascertained with any degree of approximation. I would exercise more caution and state that in the absence of further evidence, an assessment of the importance of these structural features, which have emerged from greatly incomplete, low-resolution structures, must be taken with a grain of salt. More on this below.

The description of the connections to microtubules on page 16 is very sketchy and largely limited in scope. I am not sure how it could be improved, but the authors could at least indicate a full list of proteins involved, including at least the SKAP-Astrin complex, XMAP215, the CLASPs, and MCAK, which at present are not even cited. A table or an extension of a figure could be used for this. In general, I don’t see that citing a tiny amount of the original literature and resorting to other people’s reviews is a very useful exercise. One may well remove this part and clearly state that it is not the focus of the review.

I have many problems with the discussion of the two models at the end of the review. In my opinion, this final section lacks critical thinking, it is confusing, and will be hard to follow for most readers.

First of all, there should be appropriate coverage of all the evidence on the stoichiometry of kinetochore assembly. For instance, the authors should cite the work of Suzuki et al. in Nature Communications on the stoichiometry of kinetochore subunits.

Second, the authors write: “On CENP-A nucleosomes assembled on 601 DNA only a single copy of CENP-C binds [117], while when assembled on yeast CEN DNA two copies of CENP-C binds [105,117]. It is tempting to speculate that some natural DNA sequences could be refractory to centromere formation because like the artificial 601 sequence they do not accommodate the assembly of the CCAN starting with CENP-C and/or CENP-N”. Earlier in the section, they also write that “the most prominent models for the chromatin connection to microtubules have generally featured a stoichiometry of two CCAN complexes per CENP-A nucleosome (Figure 6, Model 1) [136]. This has been supported by the sedimentation behavior of reconstituted particles containing with a large subset of recombinant CCAN components [111].” These sedimentation

experiments were carried out with nucleosomes build on the 601 sequence. Does that fit the authors' narration?

Third, it is fine to discuss how the choice of DNA may influence the biochemistry of the system, but the discussion should focus on more impending questions. First, the authors should clarify whether the structure of the complex by the Barford laboratory is a reasonable account of what we know of the yeast kinetochore in the first place. How sensible is it to assume that CENP-N occupies a different position in the yeast structure relative to that of humans? How strong is the conservation of interfaces that would support functional similarities or differences? What could have gone wrong in structure determination, if anything at all (e.g. what fraction of particles out of the total were used in the reconstructions)? What are the implications of removing half of CENP-N before mitotic entry? How can the structure accommodate such a large structural change given what has been discussed before on the foundational role of CENP-N for the stability of the inner kinetochore? How could these models be tested? Importantly: an observation is not a model. Models have to try to explain something, else what are they good for? As far as model 2 is concerned, there are some clear predictions, including that the stoichiometry of other inner and outer kinetochore subunits will most likely change if half of CENP-N falls off. Has this been observed, does it fit what we know (check the Suzuki et al. paper, for instance)? Just to clarify how problematic model 2 is, the authors should consider that in *S. cerevisiae* Cnn1/CENP-T is not essential. Model 2 predicts that in cells lacking Cnn1 there will be a single functional Ndc80 tetramer (the one bound to the single Mif2/CENP-C). How realistic is that? What are the implications regarding what we know about cooperativity of NDC80 binding to microtubules? These are the type of implications that a review should discuss if it aims to have a durable impact on a field.

Finally, a note on figures. Fig. 4 shows almost everything in orange. The authors should use at least some shades of orange-reddish colors to distinguish the different chains, otherwise these diagrams won't be useful. That the orientation of Fig. 4f is different from Fig. 4 g-j is also sub-optimal. The final model in Fig. 6 could be greatly improved. What is the evidence of a "solid" connection from Ndc80 to the nucleosome, without any unstructured connecting parts where CENP-C and CENP-T? A schematic of CENP-C and CENP-T with their interacting motifs would be helpful. And a fair representation of the stoichiometries is essential.

Other points in no specific order

Incomplete sentence: "one CCAN to ultimately up to four Ndc80"

"The C-terminus ordered region contains a coiled coil followed by paired RWD domains..." "C-terminus" is a name, the adjective is C-terminal.

"Knl1 also contains multiple binding sites for proteins that associate with the outer kinetochore, including ZWINT" ZWINT is a core component of the 2-subunit KNL1 complex. It should be included in the scheme in Figure 1.

"reconstituted particles containing with a large subset of recombinant CCAN components"

Why are references coming with all authors or only the first author?

Tanaka K, Chang HL, Kagami A, Watanabe Y. 2009 CENP-C functions as a scaffold for effectors with essential kinetochore functions in mitosis and meiosis. *Dev. Cell* 17, 334– 343. (doi:10.1016/j.devcel.2009.08.004)

Xiao H et al. 2017 Molecular basis of CENP-C association with the CENP-A nucleosome at yeast centromeres. *Genes Dev.* 31, 1958–1972. (doi:10.1101/gad.304782.117)

The description of centromere sequences is rather sketchy. A diagram of the human centromere sequence would also help (i.e. a more detailed schematic to replace Fig. 2A).

“New incorporation of CENP-A happens in G1, after mitotic exit, whereas new H3 is incorporated during DNA replication [82,85]”. Here the authors should cite the paper by Elaine Dunleavy and Gary Karpen as well (Dunleavy et al., 2011 – Nucleus).

For the sections CENP-C and CENP-LN , CENP-HIKM, CENP-TWSX , CENP-OPQUR, it would be helpful to add more detailed figures of the complexes described here. There is a lot of descriptive information that will not be digested easily by non-structural biologists (and a review such this, should be a useful compendium for them). For example, Figure 4 could be expanded to have single cartoons of the complexes, clearly labelled to visually elucidate the corresponding text.

Author's Response to Decision Letter for (RSOB-20-0051.R0)

See Appendix A.

Decision letter (RSOB-20-0051.R1)

19-May-2020

Dear Dr Black

We are pleased to inform you that your manuscript entitled "The Centromere Comes into Focus –

From CENP-A Nucleosomes to Kinetochores Connections with the Spindle" has been accepted by the Editor for publication in Open Biology.

Sincerely,

The Open Biology Team
mailto:openbiology@royalsociety.org

Appendix A

Reviewer(s)' Comments to Author(s) and **author responses**:

Note: We have added numbering to Reviewer comments to more easily refer to them within this document.

Referee: 1

Comments to the Author(s)

1. This manuscript by Kixmoeller et al., does a great job of integrating all available molecular and structural level information on molecular players involved in centromere maintenance and kinetochore assembly, fundamental processes essential for cell and organismal viability. I am confident that this review article will be a valuable resource to anyone interested in understanding the fundamental inner workings of processes controlling cell survival in general and cell division in particular. This article will also be appreciated by structural biologists interested in understanding the structural basis of macromolecular assemblies and function.

Overall the manuscript is well written. However, I have a few suggestions that the authors might want to consider to make it even more effective.

We thank the Reviewer for the high assessment of our review and the suggestions for improving it. As detailed, below, we have made many changes in response.

2. The following works are not mentioned/discussed and need to be included to make this manuscript complete.

- Recent structural work on Drosophila Cal1 interaction with CENP-A(Cid)/H4 and CENP-C cupin domain interaction (Medina-Pritchard et al., EMBO J 2020) needs to be discussed both in the context of CENP-A/H4 recognition and centromere targeting.
- Where Ndc80 phosphorylation is discussed, Ndc80 phosphorylation by Aurora A kinase also needs to be mentioned (DeLuca et al., JCB 2018)
- Where the Ska complex is discussed, structural works on the Ska1 microtubule binding domains are not mentioned or cited – (Abad et al., Nature Communications 2014; Monda et al., Current Biology 2017). The work by Abad et al., provided the structural basis for how the Ska complex (unlike the Ndc80 complex) interacts with polymerising and depolymerising microtubules in an indiscriminative manner.

We added each of these.

3. Manuscript will benefit by careful re-reading to look for grammatic errors, format consistency and convoluted sentences.

Examples:

Page 2, Introduction, first paragraph, line 3 - ...replicated and segregated during every cell cycle ('during' missing);

Page 3, Paragraph 3, line 2 – where Histone H3 is introduce, better to explicitly introduce as Histone H3 rather than H3;

Page 3, Paragraph 3, line 7 – CCAN – Constitutive Centromere ... (not Constitute);

Page 3, last paragraph, line 2 - ..and in organisms (I guess, it has to be 'model organisms');
Page 4, last sentence might benefit from rephrasing;
Page 7, second paragraph, HJURP is sufficient *to* incorporation.....; (sufficient for?)
Page 11, paragraph 3, line 5 – superscript missing for CENP-I ;
Page 11, paragraph 3, line 5 from bottom – ‘...binding cite...’ – it has to be 'binding site'
Page 15, paragraph 1, line 7 – rephrasing the sentence starting ‘Supporting the importance...’ might help;
Page 15, penultimate sentence – ‘...CCAN, making it possible for one CCAN to ultimately bind up to four...’ (I guess ‘bind’ missing)
Several places, where proximity of the domain to the C or N terminus is described ‘terminal’ is better suited than ‘terminus’ – for ex: The C-terminal region or the C-terminal domain.
Page 17, paragraph 3, line 6 – ‘...reconstituted particles containing *with* a large subset ...’
Page 17, line 4 from bottom, ‘...DNA leads to more faithful assembly...’ – I am not convinced that the use of ‘faithful’ here is appropriate.
Page 18, line 5, rephrasing the sentence starting ‘On CENP-A nucleosomes...’ might benefit.

We apologize for the initial errors and thank the reviewer for taking the time to point these out. We fixed all of these (and others we identified).

4. Where CENP-A CATD introduced, it would help to include a panel showing the sequence alignment with secondary structure elements and the domain

This is added as the new Figure 2c.

5. It would help to use arrows in the figures wherever possible, to highlight features that the readers are expected to focus. This will particularly help those general readers who are not very familiar looking at protein structures. Labeling domains (such as the CH domain, RWD domain) in the figures will also help. It would also help to include the domain architecture of CENP-C in the figure, where it is discussed.

We thank the reviewer for encouraging us to highlight features in this way. We have updated Figures 3, 5, and 6 accordingly.

6. Inner and outer kinetochore subcomplexes are shown in similar colors – it would help if contrasting colours can be chosen.

The bright pink and deep orange are contrasting, and we’ve now made new panels and adjusted other ones in Figures 5 and 6 to make individual CCAN and kinetochore components in different shades.

7. In the figure legends, where possible organism name needs to be included – if the structure comes from pombe, cerevisiae or human.

These are now added to the legends.

Referee: 2

Comments to the Author(s)

This review by Kixmoeller and coworkers focuses on the kinetochore, a large macromolecular complex that bridges chromosome and microtubules during mitosis to the segregation of chromosomes in the daughter cells. There has been considerable recent progress in this area, fueled by efforts of biochemical reconstitution and structural analysis. A review on the topic is therefore timely and has the potential to occupy an empty niche.

1. The review aims to be comprehensive. This is not easy, because kinetochores are very large structures. In its present form, the review focuses more closely on the so-called CCAN, the inner kinetochore complex. The outer kinetochore and its control of microtubule binding is treated rather superficially and one wonders if it should be covered at all at this superficial level. It leaves a feeling of incompleteness.

We have addressed this by making clear (with revisions to the title and introduction) our motivation for the balance we tried to strike in writing a review on this fundamental biological process. One that we know is the subject of books that still struggle to cover all the important aspects. We hope that the reviewer will understand why a review covering the areas we have covered and with the balance in scope is valuable to the field and as a resource for generalists interested in these aspects of the chromosome segregation mechanism. In addition, and in specific relation to some other points (including point #16, below), we have tried to direct the reader to other important kinetochore proteins and where they can find some of the foundational research papers (and some reviews, where appropriate).

2. With concern to CCAN, on the other hand, the review covers individual interactions between kinetochore protein complexes in some detail, therefore making it a good summary for people approaching the field. On the other hand, the figures lack detail on the individual protein complexes, and are difficult to read. The manuscript would benefit considerably if figures are improved.

We have made improvements to the figure set, with improvements/added details to the current Figures 2, 3, 6, and 7 and a new Figure 5.

3. Another issue that the authors should consider is that the review falls short of making critical sense of the implications of two models being proposed based on current structural information. This is detailed below. In my opinion, models should try to explain something, i.e. should be clearly distinct from the observations that support them, and it should be clear what their value is. This is not what happened here.

We thank the Reviewer for pushing us for making this important part of the Review stronger. In the revision, we have added an expanded discussion (see pgs. 17-20) of the two models. We hope it is clear to the Reviewer now.

4. Finally, there are problems with the cited literature, some of which are listed below.

We apologize for not getting these correct the first time and thank the reviewer for making the effort to guide us in each specific instance. We have also gone through and corrected other errors that were in the original submission. The revision is certainly much improved by these additions.

5. In summary, I praise the authors for their effort. At the same time, I feel rather lukewarm about this review and what it tries to achieve. I hope that the following comment will direct the authors towards assembling an improved review.

We thank the Reviewer for the praise, and in response to the helpful critiques we think we have assembled an improved review.

Specific points

6. The choice of cited literature is absolutely crucial in a review and conveys a sense of how authoritative and influential the review is going to be. Here it is “O.K-ish”. It could be improved to remove inaccuracies and inconsistencies. Here are a few examples, but the problem is more pervasive.

“Structures are available for each individual component bound or in combination (Figure 3b-e) [67,68,72,108].” The choice of literature supporting this statement is incomplete. The authors should specifically mention the work of Chittori et al. paper here and also cite Tian et al.: Tian T, Li X, Liu Y, Wang C, Liu X, Bi G, Zhang X, Yao X, Zhou ZH, Zang J. Molecular basis for CENP-N recognition of CENP-A nucleosome on the human kinetochore. Cell Res. 2018 Mar;28(3):374-378. doi: 10.1038/cr.2018.13.

We have added these. For the Tian et al, reference, we were aware of that paper, but we don't know how the Editors of Open Biology feel about their journal citing papers in the category of that paper (it appears to be a “Letter to the Editor” and it is not clear that it was peer reviewed). We ask for Editorial guidance for whether or not it would be better to now remove that particular citation.

7. “Other crystallographic studies have reported the structures of CENP-M (Figure 4c) [111]” Reference 111, Weir et al. 2016, does not report the structure of CENP-M. That was rather described in Basilico et al. 2014 (reference 126).

Now fixed.

8. “Both CENP-C and CENP-N are required to recruit all other CCAN components [23,63,64,65,108,109,118,121,122].” Why is reference 121 part of this group?

Now removed.

9. “The complex is composed of two heterodimers, Ndc80-Nuf2, and Spc24-Spc25 [143,149,150].” These are important references but they do not make that point, which is

rather the focus of references 151-153 cited two lines later. What about John Kilmartin's work? He was the one showing Ndc80 is a tetramer.

We have fixed/added these citations, as suggested.

10. "Supporting the importance of cooperative binding is evidence that, unlike individual Ndc80 complexes in solution, Ndc80 complexes immobilized at high concentration on beads can create load-bearing attachments to depolymerizing microtubules [2,26,159,160]." Here the authors cite three reviews and a single research paper. The reviews do not focus at all on the issue of cooperative binding. The research paper has one experiment on this, but the issue has been the subject of more recent work (e.g from the labs of De Luca, Grishchuk, and Musacchio). If the authors do not want to cite original literature, they could elect to cite the 2020 Wimbish and DeLuca excellent review that is entirely dedicated to this topic.

We have addressed this by adding several of the original research papers.

11. "The 4-subunit Mis12 complex contains Mis12, Pmf1, Nsl1, and Dsn1 [162]. The budding yeast equivalent of this complex is the MIND (or Mtw1) complex [163]." Again, this is an incorrect choice of literature in this context. Both references describe analyses of the human complex, not of the yeast complex. The structure of the yeast complex was described by Dimitrova et al., reference 166.

Now fixed.

12. "The Mis12 complex interacts with both CENP-C and CENP-T, serving as an important interaction hub between the KMN assembly and the inner kinetochore [164]." Reference 164 describes the interaction of MIS12 with CENP-C, but the interaction with CENP-T was described by Gascoigne et al. in humans and Malvezzi et al. in yeast.

Now fixed.

13. "This retraction of the kinetochore involves the release of the dynein/dynactin and RZZ complexes along with the central SAC complex, halting SAC signaling at that kinetochore [180]." Reference 180 is excellent but it focuses on Spindly, a protein that is not even included in the authors' list of corona component. The role of RZZ in checkpoint signaling was established in the first decade of this millennium in the Salmon and Karess labs.

Now fixed.

14. There are several of these shortcomings and I would therefore urge the authors to go through the review and try to remove these inaccuracies and inconsistencies.

We thank the Reviewer for the encouragement to do this. We have done this, leading to many changes and additions throughout.

15. Page 6. The second half of this page is devoted to the description of the loose ends of the CENP-A nucleosome, super-helical bulges, and to whatever else is different relative to H3 nucleosomes, stressing that the structural changes are important for CCAN assembly. All this comes across as a reflection of the authors' emphasis on their own work, and they can hardly be blamed for it as this is their review. My personal opinion, however, is that in the absence of structures of the human CCAN-CENP-A nucleosome complex, the importance of these effects cannot be ascertained with any degree of approximation. I would exercise more caution and state that in the absence of further evidence, an assessment of the importance of these structural features, which have emerged from greatly incomplete, low-resolution structures, must be taken with a grain of salt. More on this below.

We now finish the paragraph in question with a statement about the importance of future work in this area. In response to this point, we also added a statement on pg. 20 about the importance of future experimentation with the assembly and structure of the CENP-A nucleosome/CCAN complex.

16. The description of the connections to microtubules on page 16 is very sketchy and largely limited in scope. I am not sure how it could be improved, but the authors could at least indicate a full list of proteins involved, including at least the SKAP-Astrin complex, XMAP215, the CLASPs, and MCAK, which at present are not even cited. A table or an extension of a figure could be used for this. In general, I don't see that citing a tiny amount of the original literature and resorting to other people's reviews is a very useful exercise. One may well remove this part and clearly state that it is not the focus of the review.

We have made the requested additions, citations, and clarifications on pgs. 16-17.

17. I have many problems with the discussion of the two models at the end of the review. In my opinion, this final section lacks critical thinking, it is confusing, and will be hard to follow for most readers.

First of all, there should be appropriate coverage of all the evidence on the stoichiometry of kinetochore assembly. For instance, the authors should cite the work of Suzuki et al. in Nature Communications on the stoichiometry of kinetochore subunits.

This is now referred to and cited on pg. 17.

18. Second, the authors write: "On CENP-A nucleosomes assembled on 601 DNA only a single copy of CENP-C binds [117], while when assembled on yeast CEN DNA two copies of CENP-C binds [105,117]. It is tempting to speculate that some natural DNA sequences could be refractory to centromere formation because like the artificial 601 sequence they do not accommodate the assembly of the CCAN starting with CENP-C and/or CENP-N". Earlier in the section, they also write that "the most prominent models for the chromatin connection to microtubules have generally featured a stoichiometry of two CCAN complexes per CENP-A nucleosome (Figure 6, Model 1) [136]. This has been supported by the sedimentation behavior of reconstituted particles containing with a large subset of recombinant CCAN

components [111].” These sedimentation experiments were carried out with nucleosomes build on the 601 sequence. Does that fit the authors’ narration?

We have overhauled this section and extended it, with a particular mention of this point on pg. 19.

19. Third, it is fine to discuss how the choice of DNA may influence the biochemistry of the system, but the discussion should focus on more impending questions. First, the authors should clarify whether the structure of the complex by the Barford laboratory is a reasonable account of what we know of the yeast kinetochore in the first place. How sensible is it to assume that CENP-N occupies a different position in the yeast structure relative to that of humans? How strong is the conservation of interfaces that would support functional similarities or differences? What could have gone wrong in structure determination, if anything at all (e.g. what fraction of particles out of the total were used in the reconstructions)? What are the implications of removing half of CENP-N before mitotic entry? How can the structure accommodate such a large structural change given what has been discussed before on the foundational role of CENP-N for the stability of the inner kinetochore? How could these models be tested? Importantly: an observation is not a model. Models have to try to explain something, else what are they good for? As far as model 2 is concerned, there are some clear predictions, including that the stoichiometry of other inner and outer kinetochore subunits will most likely change if half of CENP-N falls off. Has this been observed, does it fit what we know (check the Suzuki et al. paper, for instance)? Just to clarify how problematic model 2 is, the authors should consider that in *S. cerevisiae* Cnn1/CENP-T is not essential. Model 2 predicts that in cells lacking Cnn1 there will be a single functional Ndc80 tetramer (the one bound to the single Mif2/CENP-C). How realistic is that? What are the implications regarding what we know about cooperativity of NDC80 binding to microtubules? These are the type of implications that a review should discuss if it aims to have a durable impact on a field.

We particularly thank the Reviewer for pushing us to improve this part. We have added extensively to the discussion of the models presented (Fig. 7 in the revised manuscript)(see new text on pgs. 17-20). Along with the clarification in the title and introduction about the focus of our review, we now also clarify what our models are meant to highlight and what they are not. Certainly every model that has been put forth has been “problematic” if one’s criteria is that all centromeres from all eukaryotes must work precisely in the same manner. We discuss that point, too (see additions on pg. 18), and considered providing even more examples (such as fruit fly where the CCAN is essentially missing despite the presence of CENP-A nucleosomes and CENP-C). But in the end, we tried to find a balance that would be interesting and helpful for both the centromere crowd and a broader readership of biologists. While we presume from this lineup of questions from the Reviewer that what we have ultimately written might not match what the Reviewer would write in a Review of their own, we hope that they find this part of our Review to contain a deeper explanation of the models put forth.

20. Finally, a note on figures. Fig. 4 shows almost everything in orange. The authors should use at least some shades of orange-reddish colors to distinguish the different chains,

otherwise these diagrams won't be useful. That the orientation of Fig. 4f is different from Fig. 4 g-j is also sub-optimal.

We addressed all of these issues in the new Figure 5.

21. The final model in Fig. 6 could be greatly improved. What is the evidence of a "solid" connection from Ndc80 to the nucleosome, without any unstructured connecting parts where CENP-C and CENP-T? A schematic of CENP-C and CENP-T with their interacting motifs would be helpful. And a fair representation of the stoichiometries is essential.

We've added the flexible extensions, and thank the Reviewer for that suggestion. As mentioned for point #19, above, that we've indicated in the text what the point of showing the models in Figure 7. The main point is not to discuss/show the stoichiometries of those components, but we address some of the proposed stoichiometries specifically in the text (on pgs. 17-18) and mention in the figure legend how putative connections to other nucleosomes by CENP-C are removed for simplicity.

Other points in no specific order

22. Incomplete sentence: "one CCAN to ultimately up to four Ndc80"

Now fixed.

23. "The C-terminus ordered region contains a coiled coil followed by paired RWD domains..." "C-terminus" is a name, the adjective is C-terminal.

Now fixed.

24. "Knl1 also contains multiple binding sites for proteins that associate with the outer kinetochore, including ZWINT" ZWINT is a core component of the 2-subunit KNL1 complex. It should be included in the scheme in Figure 1.

Now done.

24. "reconstituted particles containing with a large subset of recombinant CCAN components"

Now fixed.

25. Why are references coming with all authors or only the first author?

Tanaka K, Chang HL, Kagami A, Watanabe Y. 2009 CENP-C functions as a scaffold for effectors with essential kinetochore functions in mitosis and meiosis. Dev. Cell 17, 334–343. (doi:10.1016/j.devcel.2009.08.004)

Xiao H et al. 2017 Molecular basis of CENP-C association with the CENP-A nucleosome at yeast centromeres. *Genes Dev.* 31, 1958–1972. (doi:10.1101/gad.304782.117)

It is our understanding that the former is the Open Biology format for papers with a smaller number of authors, and the latter is the format for papers with more than 10 authors.

26. The description of centromere sequences is rather sketchy. A diagram of the human centromere sequence would also help (i.e. a more detailed schematic to replace Fig. 2A).

This is now added.

27. “New incorporation of CENP-A happens in G1, after mitotic exit, whereas new H3 is incorporated during DNA replication [82,85]”. Here the authors should cite the paper by Elaine Dunleavy and Gary Karpen as well (Dunleavy et al., 2011 – Nucleus).

Now added.

28. For the sections CENP-C and CENP-LN , CENP-HIKM, CENP-TWSX , CENP-OPQUR, it would be helpful to add more detailed figures of the complexes described here. There is a lot of descriptive information that will not be digested easily by non-structural biologists (and a review such this, should be a useful compendium for them). For example, Figure 4 could be expanded to have single cartoons of the complexes, clearly labelled to visually elucidate the corresponding text.

As alluded to in our response to #20, above, the new Figure 5 adds ribbon diagrams and labels to connect better with the text.